# A short introduction to topological quantum computation

**Ville T. Lahtinen[1] and Jiannis K. Pachos[2]⋆**

**1** Freie Universität Berlin, Arnimallee 14, 14195 Berlin, Germany
**2** School of Physics and Astronomy, University of Leeds, Leeds, LS2 9JT, United Kingdom

⋆ j.k.pachos@leeds.ac.uk

## Abstract

**This review presents an entry-level introduction to topological quantum computation – quantum computing with anyons. We introduce anyons at the system-independent level of anyon models and discuss the key concepts of protected fusion spaces and statistical quantum evolutions for encoding and processing quantum information. Both the encoding and the processing are inherently resilient against errors due to their topological nature, thus promising to overcome one of the main obstacles for the realisation of quantum computers. We outline the general steps of topological quantum computation, as well as discuss various challenges faced by it. We also review the literature on condensed matter systems where anyons can emerge. Finally, the appearance of anyons and employing them for quantum computation is demonstrated in the context of a simple microscopic model – the topological superconducting nanowire – that describes the low-energy physics of several experimentally relevant settings. This model supports localised Majorana zero modes that are the simplest and the experimentally most tractable types of anyons that are needed to perform topological quantum computation.**

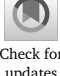
# 1 Introduction

Topological quantum computation is an approach to storing and manipulating quantum information that employs exotic quasiparticles, called anyons. Anyons are interesting on their own right in fundamental physics, as they generalise the statistics of the commonly known bosons and fermions. Due to this exotic statistical behaviour, they exhibit non-trivial quantum evolutions that are described by topology, i.e. they are abstracted from local geometrical details. When anyons are used to encode and process quantum information, this topological behaviour provides a much desired resilience against control errors and perturbations. To be more precise, the presence of certain kinds of anyons gives rise to a degenerate decoherence-free subspace, in which the state can only be evolved by moving the anyons adiabatically around each other. While from the first sight anyons appear to be an over-complicated method for performing quantum computation, they are profoundly linked to quantum error correction [1], the algorithmic means we have in dealing with errors during quantum computation. In a sense, anyonic quantum computers implement quantum error-correction at the hardware level, thus becoming resilient to control errors and erroneous perturbations. This has augmented topological quantum computation from a niche field of research to a methodology that permeates much of the research efforts in realising fault-tolerant quantum computation.

In this review we present a non-technical introduction to anyons and to the framework for performing fault-tolerant quantum computation with them. The emphasis will be on the key properties that define anyons and how they enable protected encoding and processing of quantum information. As anyons can emerge in numerous microscopically distinct systems, we discuss these concepts primarily at a platform-independent level of anyon models and provide an extensive list of references for the interested reader to go deeper. Our aim is to provide a clear and concise introduction to the underlying principles of topological quantum computation without expertise in condensed matter theory. When condensed matter concepts are needed, we introduce them in a heuristic level to give the reader a general understanding without referring to the mathematical details. In doing so, we aim this review to be accessible to anyone with a solid undergraduate understanding of quantum mechanics and the basics of quantum information.

While several reviews have already been written on the topic, we aim this review to serve as

an accessible starting point. Topological quantum computation with fractional quantum Hall States is reviewed extensively by Nayak et al. [2], while Das Sarma et al. focus on quantum computation with Majorana zero modes [3]. The book by Pachos can be viewed as an extended version of the present review that goes deeper into the condensed matter topics [4], while the book by Wang focuses on the more mathematical aspects and their connections to knot theory [5]. The reader may also find useful the lecture notes by Roy and DiVincenzo [6] and the classic lecture notes by Preskill [7]. This review concerns only topological quantum computation where both the encoding and processing of quantum information is topologically protected. Anyons have applications also to quantum memories and quantum error correction, i.e. when only the encoding is topologically protected. For reviews on topological quantum memories, we refer the interested reader to the reviews by Terhal [8] and Brown et al. [9].

This review is structured as follows. We begin in Section 1.1 by describing at a heuristic level why topology can increase fault-tolerance and why the dimension of space is paramount when looking for systems that support anyons. In Section 2 we discuss the different types of topological order and the conditions under which they can support anyons. An extensive list of known systems of anyons is provided and we also outline how the defining properties of anyons manifest themselves in microscopic systems. In Section 3 we turn to the system independent discussion of anyon models and describe how a minimal set of data captures all the dynamics associated with a given anyon model. As examples we consider both Fibonacci anyons (what we would like to have for topological quantum computation) and Ising anyons (what we have so far). Section 4 is the core of the review where we explicitly discuss how Ising anyons can be used to encode and process quantum information in a topologically protected manner, while in Section 5 we illustrate how such quantum computation could be carried out in a specific microscopic system. As an example we employ superconducting nanowire arrays that support Majorana zero modes and that are currently the experimentally most promising direction. We conclude with Section 6.

## 1.1 Topology, stability and anyons

In mathematics, topology is the study of the global properties of manifolds that are insensitive to local smooth deformations. The overused, but still illustrative example is the topological equivalence between a donut and a coffee cup. Regardless of the local details that give them rather different everyday practicalities, both are mathematically described by genus one manifolds meaning that there is a single hole in both. Small smooth deformations, such as taking a bite on the side of the donut or chipping away a piece of the cup will change the object locally, but the topology remains unchanged. Only global violent deformations, such as cutting the donut in half or breaking the cup handle, will change the topology by removing the hole.

However, in real world small deformations matter. Quite spectacular salesmanship is required to sell a donut from which someone has already taken a bite. Something similar occurs also in quantum mechanics. To store and evolve a pure quantum state coherently, one must take exceptional care that no outside noise interferes and that the evolution is precisely the desired one. This is the key fundamental challenge in quantum computation: to robustly store quantum states for long times and evolve them according to specific quantum gates. Were quantum information encoded in topological properties of matter, and were the quantum gates dependent only on the topology of the evolutions, then both should be inherently protected from local perturbations. Such topological quantum computation would exhibit inherent hardware-level stability that ideally would make elaborate schemes of quantum error-correction redundant.

This idea was first floated by Kitaev in connection to surface codes for quantum error correction [1,10]. He realized that certain codes could be viewed as spin lattice models, where the elementary excitations are *anyons* – quasiparticles with statistics interpolating between

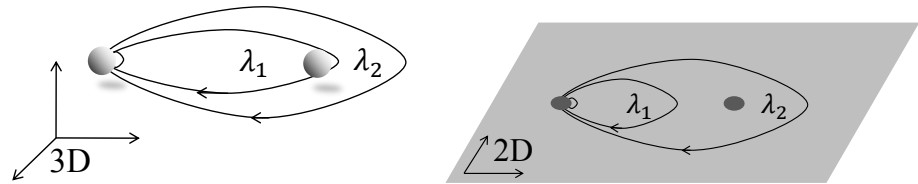

Figure 1: Exchange statistics in 2D vs. 3D. In 3D the path $\lambda_2$ describing two particle exchanges is continuously deformable to $\lambda_1$ by taking it behind or front of the right-most particle, and in turn $\lambda_1$ is contractible to a point. Hence, all the paths have the same topology and thus correspond to the same statistical quantum evolution. In 2D, however, the paths $\lambda_2$ and $\lambda_1$ are topologically inequivalent since $\lambda_2$ can not be deformed through the right-most particle, while $\lambda_1$ is still contractible to a point. Hence, the paths now have different topology and different statistical quantum evolutions can be assigned to each.

those of bosons and fermions [11]. By manipulating these excitations, quantum states could be encoded in the global properties of the system and manipulated by transporting the anyons along non-contractible paths. The local nature of the paths would be irrelevant – any two paths that were topologically equivalent implemented the same quantum gate. This insight put the study of anyons at the center of topological quantum computation. Importantly, it gave significant renewed incentive to condensed-matter physicists to look for realistic systems that could give rise to them.

The reason why anyons can exist in general can be traced back to the simple, but far reaching realization that local physics should remain unchanged when two identical particles are exchanged. In three spatial dimensions (3D) this dictates that only bosons and fermions can exist as point-like particles. A wave function describing the system of either types of particles acquires a $+1$ or a $-1$ phase, respectively, whenever they are exchanged. However, when one goes down to two spatial dimensions (2D), a much richer variety of statistical behaviour is allowed. In addition to bosonic and fermionic exchange statistics, arbitrary phase factors, or even non-trivial unitary evolutions, can be obtained when two particles are exchanged. This fundamental difference between 2D and 3D arises due to the different topology of space-time evolutions of point-like particles. Consider the exchange processes of two particles illustrated in Figure 1. In 3D the path $\lambda_2$ drawn by the encircling particle is always continuously deformable to the path $\lambda_1$ that does not encircle the other particle (the path can be deformed to pass behind the other particle). This loop, in turn, is fully contractible to a point, which means that the wave function of the system must satisfy

$$\text{3D}: \quad |\Psi(\lambda_2)\rangle = |\Psi(\lambda_1)\rangle = |\Psi(0)\rangle. \quad (1)$$

As one particle encircles the other twice, the evolution of the system can be represented by the exchange operator $R$ such that $|\Psi(\lambda_2)\rangle = R^2|\Psi(0)\rangle$. The contractibility of the loop requires that $R^2 = 1$, which has only the solutions $R = \pm 1$ that correspond to the exchange statistics of either bosons or fermions. This means that the order and the orientation of the exchanges are not relevant and the statistics of point-like particles in 3D are mathematically described by the permutation group.

This contrasts with 2D, where the path $\lambda_2$ is no longer continuously deformable (the path is not allowed to pass through the encircled particle) to the fully contractible path $\lambda_1$. This means that the final state $|\Psi(\lambda_2)\rangle$ no longer needs to equal the initial state $|\Psi(0)\rangle$

$$\text{2D}: \quad |\Psi(\lambda_2)\rangle \neq |\Psi(\lambda_1)\rangle = |\Psi(0)\rangle. \quad (2)$$

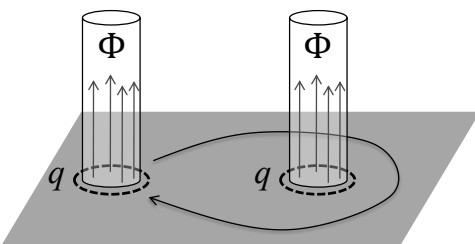

Figure 2: Toy model for anyons as charge-flux composites where magnetic flux $\Phi$ confined to a tube that is encircled by a ring of electric charge $q$. When one such composite object moves around the other, its charge (flux) circulates the flux (charge) of the other anyon. The Aharonov-Bohm effect gives rise to the complex phase $e^{2iq\Phi}$, which describes the mutual statistics of the composite objects. If $2q\Phi$ is not an integer multiple of $2\pi$, the composite objects are Abelian anyons.

Hence the exchange operator $R$ is no longer constrained to square to identity either. Instead, it can be represented by a complex phase, or even a unitary matrix. In the first case the anyons are called *Abelian anyons* due to their exchange operators commuting, while in the latter case the anyons are referred to as *non-Abelian anyons*. Since one no longer demands $R = R^{-1}$, the order and orientation of the exchanges are physical and the only constraints on the exchange operator $R$ are given by consistency conditions for distinct evolutions. These derive from a mathematical structure known as the braid group, which describes all topologically distinct evolutions of point-like particles in two spatial dimensions. It is this description of the 2D statistics by the braid group, instead of the permutation group, that allows anyons to exist.

While the possibility of something to exist is not equivalent to it actually existing, this simple analysis gives the key hint for where to look for anyons – systems that are 2D. As we are living in a 3D world, no genuine 2D systems exist. Nevertheless, many systems can be constrained to exhibit effective 2D behavior, such as electron gases at 2D interfaces of 3D materials, isolated sheets of atoms such as graphene or 2D optical lattices of cold atoms. One should keep in mind though that 2D only enables anyons to exist, but by no means guarantees that. In fact, the emergence of anyons requires further special conditions, that can be illustrated using an intuitive toy model for anyons [12]. Consider a composite particle that consists of a magnetic flux $\Phi$ confined inside a small solenoid and a ring of electric charge $q$ around it, as illustrated in Figure 2. If one such quasiparticle encircles the other, then its charge $q$ goes around the flux $\Phi$, and vice versa. Due to the celebrated Aharonov-Bohm effect [13], the wave function of the system acquires a phase factor $e^{2iq\Phi}$, even if there is no direct interaction between the quasiparticles. Since all the magnetic flux is confined to the solenoid, this phase factor does not depend in the local details of the path, which makes it topological in nature. Thus the wave function will evolve exactly in the same way as a system of two particles with exchange statistics described by $R = e^{iq\Phi}$. If

$$q\Phi \neq \pi n, \qquad n = 1, 2, \dots , \qquad (3)$$

then the composite particles are Abelian anyons. For instance, if the flux is given by half of a flux quantum $\Phi = \pi$, then any fractionalized charge $q \neq n$ (in units of the electron charge) makes the flux-charge composites anyons. While this is a toy picture, it hints of an intimate connection between anyons and fractionalisation.

Whether fractionalized quasiparticles emerge in a given microscopic system is a model specific question that has no universal answer. Most of the known systems require strong interactions between the elementary particles, such as electrons. However, also defects in

certain non-interacting states can give rise to anyons, as is the case with superconducting nanowires that we discuss in Section 5. We now turn to discuss different types of 2D topological states of matter and review the literature on those that either support or are proposed to support anyons.

## 2 Topological order and anyons in condensed matter systems

Nowadays it is impossible to talk of distinct phases of matter without talking about topology. Topological insulators and superconductors, Weyl or Dirac semi-metals, both integer and fractional quantum Hall states and spin liquids are all instances of topological states of matter. Without going into the details, mathematically every topological state of matter is characterized by some topological invariant, that can be calculated from the ground state wavefunction and that takes different values in different states. However, when it comes to supporting anyons, not all topological states are equal. In this subsection we give a brief overview of distinct topological states of matter, the general features required for them to support anyons and list those for which there is either theoretical or experimental evidence. We conclude the section by outlining how the defining properties of anyons manifest themselves in such microscopic systems.

Topological states of matter come in two broad distinct classes. The first class consists of states that are topological only given that some protecting symmetry is respected. If the symmetry is broken, the states immediately become trivial. Hence, such states are commonly referred to as *symmetry-protected topological (SPT) states*. They include all integer quantum Hall states and topological insulators and superconductors (for reviews we refer to [14, 15]), that have been classified based on the fundamental symmetries of time-reversal and charge-conjugation [16] as well as based on symmetries arising from the underlying crystal lattice [17–20]. All these are systems of non-interacting fermions, but SPT states can also emerge in bosonic systems, such as spin chains or lattice models, and in the presence of interactions [21–23]. While SPT states exhibit interesting phenomena, such as protected surface currents even if the bulk is an insulator, they *do not* support anyons as intrinsic quasiparticle excitions. However, there is an exception to this rule if the systems are allowed to have *defects*, such as domain walls between different states of matter, vortices in a superconductor or lattice dislocations. These may bind localized zero energy modes that can behave as anyons. In particular, in topological superconductors localized zero energy modes are described by Majorana (real) fermions that can be viewed as fractionalized halves of complex fermion modes [27, 28, 54]. For all practical purposes Majorana zero modes behave like non-Abelian anyons, but they are also the most complex kind of anyonic quasiparticles that can emerge in SPT states of free fermions. Unfortunately, they are not universal for quantum computation by purely topological means, but as the most experimentally accessible anyons, they have been the subject of intense research efforts. We review briefly these developments below and in Section 5 employ them to illustrate how topological quantum computation could be carried out in an array of topological superconducting nanowires.

The second class consists of states with *intrinsic topological order* that does not require any symmetries to be present. This class includes the strongly interacting fractional quantum Hall states and spin liquids. These states always support different kinds of anyons as intrinsic quasiparticle excitations (no defects are required and anyons beyond Majorana zero modes are in principle available), some of which are universal for quantum computation by purely topological means. Unlike SPT states, ground states of intrinsically topologically ordered states also exhibit long-range entanglement that gives rise to topological entanglement entropy [24–26] and ground state degeneracy that depends on the topology of the manifold the system is de-

fined on [29]. While these concepts are not directly related to topological quantum computation, they are of paramount importance in identifying the presence of intrinsic topological order in different models and we review them briefly in Section 2.2.

## 2.1 Topological states that support anyons

As discussed above, anyons require either states with intrinsic topological order or SPT states with some kind of defects. Regarding the latter, of particular interest are topological superconductors that support Majorana zero modes. To briefly review the literature on condensed matter systems that support anyons of interest to topological quantum computation, we discuss them under three broad classes: (i) Strongly correlated electron gases under strong magnetic fields that support fractional quantum Hall states, (ii) collective states of strongly interacting spins giving rise to spin liquid states and (iii) topological superconductors and their engineered variants in superconductor-topological insulator / semi-conductor heterostructures.

### (i) Fractional quantum Hall states

Fractional quantum Hall (FQH) states occur when very cold electron gases are subjected to high magnetic fields. In such states the electrons localize and form so called Landau levels, which makes the system a highly degenerate insulator. Since the electron motion is frozen out, interactions between the electrons become significant. Due to the interactions the gapped ground states at different magnetic fields can be described by a non-integer filling fraction $\nu$ – the number of electrons per flux quantum, i.e. fractionalization occurs. While the bulk of the 2D system is insulating, experimentally such states are most easily detected by measuring the Hall conductance as the function of applied magnetic field. This exhibits characteristic plateaus corresponding to different filling fractions [30,31]. Every plateau is a distinct phase of intrinsic topological matter, characterized by the quantized Hall conductance that is proportional to the topological invariant characterizing the state [32].

The nature of these intrinsically topologically ordered states and the anyons they support is understood via trial wave functions. Such wave functions were first proposed by Laughlin to predict that the $\nu = 1/3$ state supports fractionalized quasiparticles that behave as Abelian anyons [33]. While the direct probing of the anyons via their exchange statistics remains elusive, the charge fractionalization has been experimentally confirmed, thus strongly supporting the existence of anyons [34, 35]. Numerous further trial wave functions have been proposed to describe other filling fractions seen in the experiments. From the point of view of quantum computation, of particular interest is the $\nu = 5/2$ case. It has been proposed that at this filling fraction the system is described by the Moore-Read state that supports the simplest non-Abelian anyons – the Ising anyons [36]. The predicted charge fractionalization has been confirmed, but again direct probing of the anyons has remained elusive [38]. For most practical purposes Ising anyons are equivalent to Majorana modes and hence they are not universal for quantum computation. For universality one needs more complex Fibonacci anyons. These are expected to emerge in the very fragile $\nu = 12/5$ filling fraction described by the Read-Rezayi state [37]. For a comprehensive account of topological quantum computation in FQH states, we refer the interested reader to [2].

It has also been proposed that FQH states can emerge in topological insulators when they are subjected to similar conditions, i.e. strong magnetic fields, strong interactions and fractional filling [39]. While such fractional topological (Chern) insulators are an intriguing alternative, it is currently unclear whether these conditions can ever be achieved in crystalline materials.

### (ii) Spin liquids

When strong Coulombic interactions localize electrons in a lattice configuration, their kinetic degrees of freedom are frozen out in a Mott insulating state. However, the electron spins still interact and can form collective states that exhibit intrinsic topological order. Such states are known as topological spin liquids [40]. The study of these systems roughly follows two distinct routes. The bottom-up route is to study common spin-spin interactions, which usually are of Heisenberg type, on distinct 2D lattices. As such systems rarely lend themselves to analytic treatment, one employs mean-field theory to understand what phases could exist [41], and relies on state-of-the-art numerics to verify the predictions. Convincing numerical evidence has been obtained that topological spin liquids that support Abelian anyons do exist on several frustrated 2D lattices (e.g. triangular or Kagome lattices) [42–46]. The top-down route is to write down idealized spin lattice models that support a given topological phase. The canonical model of this type is Kitaev's honeycomb model [47] that supports Ising anyons akin to the Moore-Read state. The model is exactly solvable, which enables the emergent anyons to be studied in detail, but the fine-tuning required for the exact solvability also means that it is unlikely to appear in nature. However, certain compounds have been proposed to be described by a perturbed version of the model [48,49], and neutron-scattering experiments have provided initial evidence for spin liquid states in these materials [50].

Generalizations of the honeycomb model exist both in 2D and 3D [51,52]. As these systems all follow the same construction, this family of models is still the only analytically tractable framework for spin liquids. In principle, spin liquids supporting any types of anyons can be defined on lattices via the quantum-double [10] or the Levin-Wen [53] construction. However, these require replacing actual spins with more generic local degrees of freedom subject to rather unphysical constraints and many-body interactions. A celebrated example of the quantum-double construction is the so called Toric Code [10]. While being an intrinsically topologically ordered states with Abelian anyons, the Toric Code is also the simplest example of a topological quantum memory that features heavily in relation to quantum memories and error correction (for reviews we refer to [8,9]).

### (iii) Topological superconductors in heterostructures

Since the seminal work by Read and Green [54], it has been known that if time-reversal symmetry is broken and the pairing in a 2D superconductor is so-called $p$-wave type, then vortices (the natural defects in superconductors) bind Majorana zero modes. While actual materials exhibiting such pairing are yet to be found (though strontium ruthenate is strongly believed to be one) it was realized that qualitatively same physics could occur when a topological insulator [55], a spin-orbit coupled semiconductor [56,57], a chain of magnetic atoms [58–60] or half-metals [61,62] is placed in the proximity of a regular $s$-wave superconductor. In other words, the combination of physics from both systems realizes effective $p$-wave superconductivity at the interface.

Following an early proposal by Kitaev [63], wires made of these materials, when deposited on top of a superconductor, were predicted to host Majorana modes at their ends, which could be probed through simple conductance measurements [64,65]. While the explicit verification of their braiding properties is yet to be carried out, several experiments on microscopically distinct setups strongly support the existence of Majorana modes [66–71]. These topological nanowire heterostructures are the most prominent candidate to experimentally test the key building blocks of topological protection and implementation of topological gates via the exchange statistics of anyons. For a comprehensive review of Majorana zero modes in solid-state systems, see e.g. [72–75].

**(iv) Quantum simulations of anyonic systems**

In addition to looking for anyons in materials, much progress has been made in simulating topological states of matter with cold atoms in optical lattices [76]. Time-reversal symmetry broken Chern insulators [77,78] have been realized and it is hoped that these systems can be pushed towards fractionalized conditions that support anyons. Proposals also exist for Kitaev's honeycomb model [79], as well as for counterparts of topological superconducting nanowires that host Majorana zero modes [80–82].

The statistical properties of Majorana zero modes can also be simulated in cavity arrays [83] and photonic quantum simulators [84,85]. These systems are only unitarily equivalent to actual topological states of matter and hence not genuinely topological in nature. However, they still realize counterparts of the protected subspaces and statistical evolutions in the presence of anyons. This makes them attractive for experimentally testing the required control to reliably manipulate quantum information in a topological-like encoding.

## 2.2 Manifestations of anyons in microscopic many-body systems

We close this section by discussing how non-Abelian anyons manifest themselves in intrinsically topologically ordered microscopic systems and highlight the similarities / differences to SPT states with defects. The common key property is the emergence of a protected degenerate subspace, where the evolution is given as a non-Abelian Berry phase when the anyons are adiabatically moved around each other. The precise structure of this protected space and the possible evolutions depend on the types of anyons and they are discussed in Section 3. We also introduce two often appearing concepts – topological entanglement entropy and topological ground state degeneracy – that are not directly relevant to topological quantum computation, but which are important diagnostic tools in identifying the presence of intrinsic topological order.

### 2.2.1 Degeneracy and Berry phases

All topological states that support anyons are insulators. By this we mean that the ground state is separated from the rest of the states in the spectrum by a spectral gap $\Delta$. When a topological system is placed on a surface of trivial topology without boundaries (we discuss the possible degeneracy that can arise from non-trivial spatial topologies in the following subsection), such as a sphere, then the ground state is unique. However, when non-Abelian anyonic quasiparticles are introduced into the system, the lowest energy state in their presence exhibits degeneracy that depends on the types of anyons, as illustrated in Figure 3. This contrasts with Abelian anyons, in whose presence the lowest energy state remains unique. This non-local degenerate manifold of states is in general exponentially degenerate in the anyon separation (degeneracy lifting anyon-anyon interactions are discussed in Section 4.4) and it is separated from any other states by the spectral gap $\Delta$.

This degenerate subspace is a collective non-local property of the non-Abelian anyons that one employs to encode quantum information in a topologically protected manner. This protection arises from the presence of an energy gap and from the non-locality. Since all the excitations in the system are massive (in SPT states anyons are massless zero modes, but the defects on which they are bound to are still massive), the energy gap suppresses spontaneous excitations in the system that could interfere with the already present anyons and thereby change the state in the degenerate subspace. The non-locality protects the state in the degenerate manifold via the realistic assumption that any noise in the system acts locally and hence may only result in local displacement of the anyons. As long as this displacement is small compared to their separation, no evolution takes place in the protected subspace. Thus

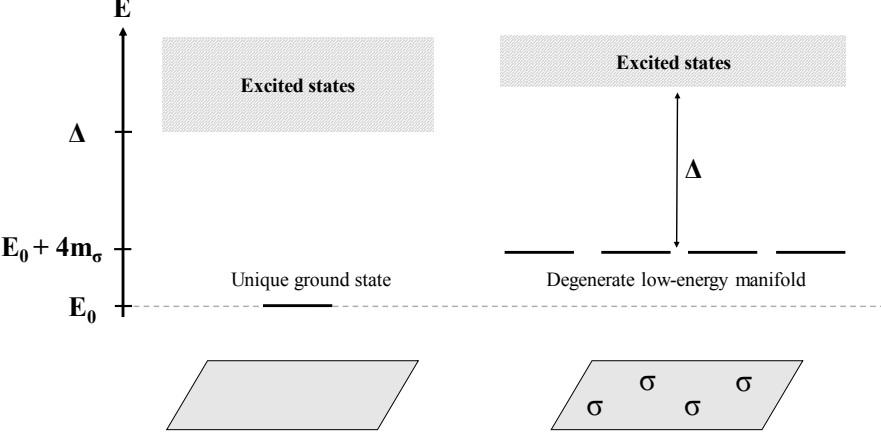

Figure 3: When an intrinsically topologically ordered state is placed on a surface of trivial topology and no anyons are present, the ground state at some energy $E_0$ is unique and separated from excited states by an energy gap $\Delta$. In the presence of non-Abelian anyons that we denote by $\sigma$, the lowest energy state exhibits degeneracy that depends on the type of anyons present (e.g. four-fold degeneracy if four $\sigma$ anyons of the Ising model are present, as we discuss soon in Section 3.4). Since anyons are massive excitations in intrinsically topologically ordered states, this degenerate manifold appears at some higher energy than the ground state in the absence of anyons (e.g. at energy $E_0 + 4m_\sigma$ when each Ising anyon has mass $m_\sigma$), but it is still separated from excited states by the spectral gap $\Delta$. Qualitatively similar picture applies also to SPT states with defects, such as a topological $p$-wave superconductor with vortices. In this case each $\sigma$ denotes a massless Majorana modes bound to a vortex, but each vortex is still carries some mass $m_\sigma$.

the degenerate low-energy manifolds in the presence of non-Abelian anyons are essentially decoherence-free subspaces.

States in the protected subspace evolve only when the anyons are transported around each other. Due to the presence of the energy gap, when this process is performed adiabatically, i.e. slowly compared to $\Delta$, the system only evolves in the non-local subspace and is given by to the statistics of the anyons. Microscopically, such evolution due to an adiabatic change in the system is given as a non-Abelian Berry phase acquired by the wave function [127–129]. Let us consider a system of $N$ non-Abelian anyons that give rise to a $D$-dimensional protected subspace. This space is spanned by $D$ degenerate many-body states given by

$$|\Psi_n(z_1, z_2, \ldots, z_N)\rangle, \qquad n = 1, 2, \ldots, D, \tag{4}$$

that depend in general on the anyon coordinates $z_j$. Let $\lambda$ be a cyclic path in $z_j$ that winds one anyon around another, as illustrated in Figure 4. If one changes the parameters $z_j$ slowly in time compared to the energy gap $\Delta$, then the transport is adiabatic and the system evolves only within the degenerate ground state manifold spanned by the states (4). This evolution is in general given by

$$|\Psi_n(z_1, z_2, \ldots, z_N)\rangle \rightarrow \sum_{m=1}^{D} \Gamma_{nm}(\lambda)|\Psi_m(z_1, z_2, \ldots, z_N)\rangle, \tag{5}$$

where the non-Abelian Berry phase is defined by

$$\Gamma(\lambda) = \mathbf{P} \exp \oint_\lambda \mathbf{A} \cdot d\mathbf{z}. \tag{6}$$

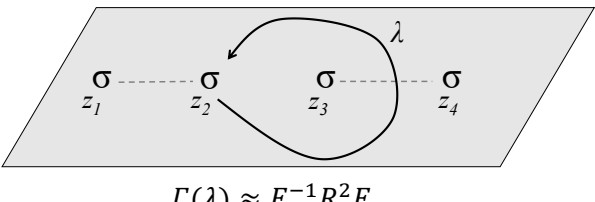

$$\Gamma(\lambda) \approx F^{-1}R^2 F$$

Figure 4: Microscopic braiding as a non-Abelian Berry phase. Consider a system where two pairs of anyons, each denoted by a $\sigma$, are created from the vacuum (the pairs are connected by dashed lines) located at positions $z_1, \ldots, z_4$. Transporting the anyon at position $z_2$ along any path $\lambda$ that encloses only the anyon at $z_3$ gives rise to non-Abelian Berry phase (6) that acts in the degenerate subspace shown in Figure 3. When the anyons correspond to Ising anyons, to be discussed in detail in Section 3.4, and the transport is adiabatic, this non-Abelian Berry phase will accurately approximate their braid matrix $F^{-1}R^2F$ given by (22).

Here **P** denotes path ordering and the components of the non-Abelian Berry connection are given by

$$(A^j)_{mn} = \langle \Psi_m(z_1, z_2, \ldots, z_N) | \frac{\partial}{\partial z_j} | \Psi_n(z_1, z_2, \ldots, z_N) \rangle. \tag{7}$$

The geometric phase due to the cyclic evolution in the coordinates $z_j$ does not depend on the time it takes to traverse the path $\lambda$ as long as it is long enough for the evolution to be adiabatic. Nor does it depend on the exact shape of the path $\lambda$ and thus the unitary $\Gamma(\lambda)$ is topological in nature. The precise form it takes depends on the types of anyons present and what their mutual anyonic statistics are. For each anyon model, there is only a finite number of possible unitary evolutions, generated by pairwise exchanges, that act in the protected non-local subspace. We discuss the different types of statistics in Section 3.

It has been verified in several microscopically distinct settings that the statistics of the anyons is indeed obtained from the adiabatic evolution of their wave functions. Analytically this has been shown for the Laughlin [130] and Moore-Read [131] fractional quantum Hall states, as well as $p$-wave superconductors [132] including heterostructure realizations of topological nanowires [133, 134]. These calculations are supported by numerics that have been used to demonstrate non-Abelian statistics in more complex fractional quantum Hall states [106, 135, 136] and in microscopic models such as Kitaev's honeycomb lattice model [109, 137] or bosonic fractional quantum Hall systems [138].

### 2.2.2 Topological degeneracy and entanglement entropy

In Section 2 we discussed the two distinct notions of topological order – SPT order and intrinsic topological order. Regarding anyons, the key difference was that the first required some kinds of defects to support, e.g., Majorana zero modes, while the latter does not. There are also two other important physical consequences of intrinsic topological order that *are absent in SPT states*.

The first property is *topological ground state degeneracy*, i.e. that the ground state in the absence of anyons is degenerate when the system is defined on a manifold of non-trivial spatial topology [29]. In other words, a state defined on a sphere (the spatial manifold has no holes) would have a unique ground state, but the same state defined on a torus (the spatial manifold has a single hole) would have a degenerate ground state space. This is a general property that applies to any intrinsic topological state regardless of the type of anyons, Abelian or non-Abelian, it supports. However, the degree of degeneracy does depend on the anyon model

and the topology of the spatial manifold and needs to be worked out on a case-by-case basis [139]. The most important consequence to quantum computation is that, given that non-trivial spatial topologies can be engineered, the degenerate ground state space can be used as a topological quantum memory. In particular, since the nature of the supported anyons is irrelevant, systems supporting relatively simpler Abelian anyons can also be employed to this end. For thorough discussion about topological quantum memories, we refer to [8, 9]. The second important application is to use the degeneracy on manifolds of distinct topology as a convenient numerical probe to identify the presence of intrinsic topological order [42–46].

The second property inherent to only intrinsic topological states is that of *topological entanglement entropy*. Any quantum state can be partitioned into two disjoint regions $A$ and $B$. The entanglement between the regions can be quantified by entanglement entropy $S = -\text{Tr}_B \rho \log \rho$, where $\rho$ is the density matrix of the full system, but the trace is taken only over the region $B$. All gapped states of matter are expected to have only short-range entanglement and thus follow the area-law scaling of entanglement entropy [140]. In other words, the entanglement between $A$ and $B$ should be proportional to the length $|\partial A|$ of the boundary between them. However, if a state exhibits intrinsic topological order, then there is also a universal constant correction to the area-law [24–26], with the entanglement entropy scaling as

$$S = \alpha |\partial A| - \gamma, \tag{8}$$

where $\alpha$ is a non-universal constant. On the other hand, $\gamma$ is a universal constant that takes a non-zero value only in the presence of intrinsic topological order.

Like the ground state degeneracy on manifolds of varying topology, $\gamma$ depends on the types of anyons supported. Extracting it by studying the entropy scaling (8) is thus another convenient numerical tool to identify the presence of intrinsic topological order. However, neither the topological ground state degeneracy nor the topological entanglement entropy are unique characteristics of anyon models – different anyon models can give rise to the same degeneracy and to the same correction to entanglement entropy. To unambiguously determine the anyon model supported by a given state, the statistics of the supported anyons can be obtained from the ground state via more sophisticated manipulations of the state [141–143].

## 3   Anyon models

From now on we adopt the perspective that anyons exist and focus on explaining what different types of anyons there can be and what their defining properties are. Furthermore, for the time being, we assume that the microscopic details of the system that give rise to the anyons can be completely neglected and the low-energy dynamics is described in terms of only the anyons. Under these assumptions the possible evolutions are limited to three simple scenarios:

1. Anyons can be created or annihilated in pairwise fashion.

2. Anyons can be fused to form other types of anyons.

3. Anyons can be exchanged adiabatically.

The formal framework capturing these properties in a unified fashion goes under the name of a *topological quantum field theory* [86]. From the point of view of topological quantum computation, most of the details of this rigorous mathematical description can be omitted. As a result, only a minimal set of data is required to specify the properties of the anyons corresponding to a particular topological quantum field theory. Here, as also often is the case in literature,

we refer to this minimal information as an *anyon model*. Such models with up to four distinct anyonic quasiparticles have been systematically classified [87]. We refer the interested reader also to the appendices of [47] for an accessible introduction to the diagrammatic notation often used when talking about anyons.

In this section we first give the general structure of anyon models to explain how a given anyon model constrains the nature of the protected subspace discussed in Section 2.2 and the possible evolutions therein. Then we illustrate these abstract concepts with two examples most relevant to quantum computation: Fibonacci anyons (universal for quantum computation, but up to now only a theoretical construction) and Ising anyons (not universal for quantum computation, but can be experimentally realized).

### 3.1 Fusion channels - Decoherence-free subspaces

To define an anyon model, one first specifies how many distinct anyons there are. For completeness, this list must include a trivial label, 1, corresponding to the vacuum with no anyons. The anyon model is then spanned by some number of particles

$$M = \{1, a, b, c, \ldots\}, \tag{9}$$

where the labels $a, b, c, \ldots$ can be viewed as topological charges carried by each anyon. As charges they must satisfy conservation rules. For anyons these are known as *fusion rules*, that take the form

$$a \times b = \sum_{c \in M} N_{ab}^c c, \tag{10}$$

where the fusion coefficients $N_{ab}^c = 0, 1, \ldots$ are non-negative integers describing the possible topological charges (fusion channels) a composite particle of $a$ and $b$ can carry ($a$ and $b$ are fused). In general $N_{ab}^c$ can be any non-negative integer, but for most physical models $N_{ab}^c = 0, 1$. If $N_{ab}^c = 0$, then fusing $a$ with $b$ can not yield $c$. If for all $a, b \in M$ there is only one $N_{ab}^c$ that is different from zero, then the fusion outcome of each pair of particles is unique and the model is Abelian. On the other hand, if for some pair of anyons $a$ and $b$ there are two or more fusion coefficients that satisfy $N_{ab}^c \neq 0$, then the model is called non-Abelian. The latter implies that the fusion of $a$ and $b$ can result in several different anyons, i.e. there is degree of freedom associated with the fusion channel. Furthermore, to conserve total topological charge every particle $a$ must have an anti-particle $b$, in the sense that $N_{ab}^1 = 1$ for some $b$. For instance, a fusion rule of the form $a \times a = 1 + b$, that we encounter below, means that $a$ is a non-Abelian anyon, as it has two possible fusion outcomes, and that it is its own anti-particle, as one of the possible fusion outcomes is the vacuum.

The key characteristic of non-Abelian anyons is that the fusion channel degrees of freedom imply a space of states spanned by different possible fusion outcomes. That is, if $a$ and $b$ can fuse to several $c \in M$, we can define orthonormal states $|ab; c\rangle$ that satisfy

$$\langle ab; c | ab; d \rangle = \delta_{cd}. \tag{11}$$

If there are $N$ distinct fusion channels in the presence of a pair of particles, the system exhibits $N$-fold degeneracy spanned by these states. We refer to this non-local space shared by the non-Abelian anyons, regardless of where they are located, as the *fusion space*. This fusion space is precisely the protected low-energy subspace discussed in Section 2.2. Under the assumption that all microscopics of the system giving rise to the anyons are decoupled from the low-energy physics, the states in the fusion space are perfectly degenerate. As it is a collective non-local property of the anyons, no local perturbation can lift the degeneracy and it is hence a decoherence-free subspace. As such it is an ideal place to non-locally encode quantum information. We stress that the fusion space arises from the distinct ways anyons *can* be fused

$$\text{(diagram)} = \sum_f \left(F^d_{abc}\right)_{ef} \text{(diagram)}$$

Figure 5: Fusion diagrams and $F$-matrices. A basis in the fusion space is given by choosing an order in which the anyons are to be fused without exchanging their positions (this results in a unitary evolution in the fusion space as discussed in the next subsection). In the case of three anyons $a$, $b$ and $c$ that are constrained to fuse to $d$, there are only two options: Either they are fused pairwise from left to right ($|(ab)c; ec; d\rangle$) or from right to left ($|a(bc); af; d\rangle$). These two bases are related by the unitary matrix $F^d_{abc}$ according to (12). The state in one basis is in general a superposition of the basis states in the other basis.

over how they *are* fused. If two anyons are actually fused and the outcome of the fusion is detected, this would correspond to performing a projective measurement in the fusion space.

The fusion space of a pair of non-Abelian anyons can not be used directly to encode a qubit though. The reason is that two states $|ab; c\rangle$ and $|ab; d\rangle$ belong to different global topological charge sectors (given by $c$ and $d$, respectively) and hence can not be superposed. Instead, one needs more than two anyons in the system, such that they can be fused in various different ways that still give the same fusion outcome when all of them are fused. The basis in such higher dimensional fusion space is given by a fusion diagram of a fixed fusion order spanned by all possible fusion outcomes. Choosing a different fusion order is equivalent to a change of basis. Like the familiar Hadamard gate that relates the $Z$- and $X$-bases of a qubit, for every anyon model there exists a set of matrices that relate a state in one basis to states in other bases. These so called $F$-matrices form part of the data of an anyon model and they are obtained as the solutions to a set of consistency conditions known as the pentagon equations [47]. We do not concern ourselves here with the explicit form of the pentagon equations, as their role is to classify different possible anyon models consistent with given fusion rules (10). For known anyon models this data can be found, e.g., in [87].

To illustrate how the $F$-matrices give structure to the fusion space, consider a case where three anyons $a$, $b$ and $c$ are constrained always to fuse to $d$ and assume that several intermediate fusion outcomes are compatible with this constraint. Then there are several fusion states belonging to the same topological charge sector $d$ that can be superposed. For three anyons there are two possible fusion diagrams corresponding to distinct bases. Either one fuses first $a$ and $b$ to give $e$, in which case the basis states are labeled by $e$ and denoted by $|(ab)c; ec; d\rangle$, or one fuses first $b$ and $c$ to give $f$, in which case the corresponding basis states are $|a(bc); af; d\rangle$. These two choices of basis must be related by a unitary matrix $F^d_{abc}$ as

$$|(ab)c; ec; d\rangle = \sum_f (F^d_{abc})_{ef} |a(bc); af; d\rangle, \tag{12}$$

where $(F^d_{abc})_{ef}$ are the matrix elements of $F^d_{abc}$, and $f$ is summed over all the anyons that $b$ and $c$ can fuse to, i.e. for which $N^f_{bc} \neq 0$. The states and the action of the $F$-matrices are most conveniently expressed in terms of fusion diagrams, as illustrated in Figure 5. Such diagrams also vividly capture the topological nature of such processes – two diagrams that can be continuously deformed into each other (i.e. no cutting or crossing of the world lines) correspond to the same state of the system.

## 3.2 Braiding anyons - Statistical quantum evolutions

The fusion space is the collective non-local property of the anyons. To evolve a quantum state in this space, the statistical properties of anyons are employed. When anyons are exchanged, or *braided* (as the process is often called due to their worldlines forming braids), the state in the fusion space undergoes a unitary evolution. Transporting the anyons along paths that do not enclose other anyons has no effect. For Abelian anyons the fusion space is one dimensional and the only possible evolution is given by a complex phase factor. This phase depends on the type of anyons and whether they are exchanged clockwise or anti-clockwise, but not on the order of the anyon exchanges. In other words, the exchange operator describing exchanging anyons $a$ and $b$ in a clockwise fashion is given by $R_{ab} = e^{i\theta_{ab}}$ for some statistical angle $0 \geq \theta_{ab} < 2\pi$. This contrasts with non-Abelian anyons for which the resulting statistical phase not only depends on the types of anyons, but also on their fusion channel $c$, i.e. the exchange is described by the operator $R_{ab}^c = e^{i\theta_{ab}^c}$. Thus, when there is fusion space degeneracy associated with different fusion channels, braiding assigns different phases to different fusion channels and depends not only on the orientation of the exchanges, but also on their order.

Given the $F$-matrices of the anyon model, the possible statistics described by the exchange operators $R_{ab}^c$ compatible with them can be obtained by solving another set of consistency equations known as hexagon equations [47, 87]. These $R$-matrices, as they are often called, describe all possible unitary evolutions that can take place in the fusion space. Consider again the case where anyons $a$, $b$ and $c$ are constrained to fuse to $d$, as in (12). Then in the basis $|(ab)c; ec; d\rangle$ a clockwise exchange of $a$ and $b$ implements the unitary

$$|(ba)c; ec; d\rangle = \sum_f R_{ab}^f \delta_{e,f} |(ab)c; ec; d\rangle, \tag{13}$$

where $f$ spans all possible fusion outcomes of $a$ and $b$ and $\delta_{e,f}$ is the Kronecker delta function. Thus for a generic state in this basis, a clockwise exchange is represented by a diagonal unitary matrix $R$ with entries $R_{ab}^f$. Were the anyons exchanged counter-clockwise, the evolution would be described by $R^\dagger$. To write down the effect of exchanging $b$ and $c$ clockwise in the same basis, one first applies the $F_{abc}^d$ to change the basis to $|a(bc); af; d\rangle$, then applies $R$, as defined above, and then returns to the original basis with $(F_{abc}^d)^{-1}$, which is given by the unitary evolution

$$|(ac)b; ec; d\rangle = (F_{abc}^d)^{-1} R (F_{abc}^d) |(ab)c; ec; d\rangle. \tag{14}$$

All unitary evolutions due to distinct exchanges of anyons can be constructed in similar fashion and they are always given by some combination of the $F$- and $R$-matrices. Again, these exchange operations are conveniently expressed diagrammatically, as illustrated in Figure 6.

Summarizing, an anyon model is most compactly specified by: (i) The fusion coefficients $N_{ab}^c$ that describe how many distinct anyons there are and how the anyons fuse, (ii) the $F$-matrices that describe the structure of the fusion space and (iii) the $R$-matrices that describe the mutual statistics of the anyons. Regardless of the microscopics that give rise to anyons in a given system, with this minimal set of data all possible states of the fusion space for arbitrary number of anyons and all the possible evolutions can be constructed. As this notation gets quickly rather cumbersome, we illustrate it with two examples that are most relevant to topological quantum computation.

## 3.3 Example 1: Fibonacci anyons

Based on their anyon model structure, Fibonacci anyons are the simplest non-Abelian anyons. There is only one anyon $\tau$ that satisfies the fusion rule

$$\tau \times \tau = 1 + \tau. \tag{15}$$

Figure 6: Exchanging anyons in different bases and the $R$-matrices. When two anyons $a$ and $b$ are exchanged in a basis where they are fused first ($|(ab)c;ec;d\rangle$), the $R$-matrix acts as a diagonal matrix that assigns a phase factor $R^e_{ab} = e^{i\theta^e_{ab}}$ depending on their fusion channel $e$. When $b$ and $c$ are exchanged, the action in the basis where $a$ and $b$ are fused first is obtained by first moving to the basis they are used first ($|a(bc);af;d\rangle$) by applying the unitary $F^d_{abc}$, then applying the diagonal $R$-matrix that assigns different phase factors to different fusion channels of $b$ and $c$ and finally returning to the original basis by applying $(F^d_{abc})^{-1}$.

In other words, $\tau$ is its own anti-particle, but two $\tau$ anyons can also behave like a single $\tau$ anyon. Repeated associative application of the fusion rule shows that the dimension of the fusion space, i.e. the number of different ways the fusion of all $\tau$ anyons can result in either total topological charge of 1 and $\tau$, grows in a rather peculiar manner

$$
\begin{aligned}
\tau \times \tau \times \tau &= 1 + 2 \cdot \tau, \\
\tau \times \tau \times \tau \times \tau &= 2 \cdot 1 + 3 \cdot \tau, \\
\tau \times \tau \times \tau \times \tau \times \tau &= 3 \cdot 1 + 5 \cdot \tau,
\end{aligned}
\tag{16}
$$

and so on. In other words the dimensionality of the fusion space in both sectors grows as the Fibonacci series, where the next number is always the sum of the two preceeding numbers (hence the name Fibonacci anyons!). This immediately points to a peculiarity of the Fibonacci fusion space. It does not have a natural tensor product structure in the sense of its dimensionality growing by a constant factor per added $\tau$ anyon.

To encode a qubit in the fusion space of Fibonacci anyons, we see from (16) that one needs three $\tau$ anyons that are constrained to fuse to a single $\tau$ particle (or equivalently, four $\tau$ anyons constrained to the total vacuum sector). A basis in this two-dimensional fusion subspace is given by the states $|(\tau\tau)\tau;1\tau;\tau\rangle$ and $|(\tau\tau)\tau;\tau\tau;\tau\rangle$, with the corresponding fusion diagrams given by Figure 5 with the corresponding label substitutions. For the Fibonacci fusion rules the $F$-matrix giving the basis transformation and the $R$-matrix describing braiding are given by

$$
F = F^\tau_{\tau\tau} = \begin{pmatrix} \phi^{-1} & \phi^{-1/2} \\ \phi^{-1/2} & -\phi^{-1} \end{pmatrix}, \qquad R = \begin{pmatrix} R^1_{\tau\tau} & 0 \\ 0 & R^\tau_{\tau\tau} \end{pmatrix} = \begin{pmatrix} e^{\frac{4\pi i}{5}} & 0 \\ 0 & e^{\frac{-3\pi i}{5}} \end{pmatrix}, \tag{17}
$$

where $\phi = (1+\sqrt{5})/2$ is the Golden Ratio (another characteristic of the Fibonacci series). For a detailed discussion on Fibonacci anyons, we refer to [88].

This seeming simplicity of the Fibonacci anyons hides something remarkable though. Namely, arbitrary unitaries can be implemented by braiding the Fibonacci anyons and hence the model is universal for quantum computation [2]! Even if $R^{10}$ equals the identity matrix,

the braid group generated by $R$ and $F^{-1}RF$ is dense in $SU(2)$ in the sense that an arbitrary unitaries can be approximated to arbitrary accuracy by only braiding the anyons. There are two serious caveats though. First, the lack of tensor product structure means that only a subspace of the full fusion space can be used to encode information. For instance, if three $\tau$ anyons are used to encode a qubit, then one would like to use altogether six anyons to encode two qubits. However, their fusion space is five-dimensional in the vacuum sector, which means that the logical qubits reside only in a subspace. The second caveat is that approximating even the simplest gates is far from straighforward. Even the $NOT$-gate requires thousands of braiding operations in very specific order [89, 90]. Several techniques exist to construct the required braids more efficiently [91–95], but the task remains challenging. Still, the fact that braiding can be employed to generate in principle arbitrary unitaries, as opposed to most other anyon models, makes Fibonacci anyons the Holy Grail for quantum computation.

While the caveats can be overcome, the most daunting thing about Fibonacci anyons is that their relative simplicityas an anyon model by no means correlates with the accessability of the microscopic systems that support them. Quite the contrary. While novel elaborate schemes to realize them have been proposed in coupled domain wall arrays of Abelian FQH states [96], the most plausible candidate is still the Read-Rezayi state that has been proposed to describe the filling fraction $\nu = 12/5$ FQH state [37]. As this state is very fragile, it remains unclear whether it can ever be realized in a laboratory. Hence, much research has focused on states hosting simpler non-Abelian anyons that might not be universal, but which still enable testing and development of topological quantum technologies. In this regard Ising anyons are of particular interest.

## 3.4 Example 2: Ising anyons

The Ising anyon model consists of two non-trivial particles $\psi$ (fermion) and $\sigma$ (anyon) satisfying the fusion rules

$$1 \times 1 = 1, \quad 1 \times \psi = \psi, \quad 1 \times \sigma = \sigma,$$
$$\psi \times \psi = 1, \quad \psi \times \sigma = \sigma,$$
$$\sigma \times \sigma = 1 + \psi.$$

The fusion rule $\psi \times \psi = 1$ implies that, when brought together, two fermions behave like there is no particle, while $\psi \times \sigma = \sigma$ implies that $\psi$ with a $\sigma$ is indistinguishable from a single $\sigma$. The non-Abelian nature of the $\sigma$ anyons is encoded in the last fusion rule, which says that two of them can behave either as the vacuum or as a fermion. Physically, the fusion rules can be understood, for instance, in the context of a topological $p$-wave superconductor [54]. There, the vacuum 1 is a condensate of Cooper pairs. The fermions $\psi$ are Bogoliubov quasiparticles that can pair into a Cooper pair and thus vanish into the vacuum. The $\sigma$ anyons, on the other hand, correspond Majorana zero modes bound to vortices. As we will explain in Section 5, a Majorana mode corresponds to a "half" of a complex fermion mode. A pair of such vortices carries thus a single non-local fermion mode, the $\psi$ particle, that can be either unoccupied (fusion channel $\sigma \times \sigma \to 1$) or occupied (fusion channel $\sigma \times \sigma \to \psi$).

The non-Abelian fusion rule for the $\sigma$ anyons implies that there is a two dimensional fusion space associated with a pair of them. The basis can be associated with the two fusion channels and denoted by $\{|\sigma\sigma; 1\rangle, |\sigma\sigma; \psi\rangle\}$. However, as these two states belong to different topological charge sectors, they can not be superposed. In order to have a non-trivial fusion space in the same charge sector, one needs to consider at least three $\sigma$ particles that can fuse to a single $\sigma$ in two distinct ways (or equivalently four $\sigma$ anyons that fuse to 1), as shown by the

repeated associative application of the fusion rules

$$
\begin{aligned}
\sigma \times \sigma \times \sigma &= 2 \cdot \sigma, \\
\sigma \times \sigma \times \sigma \times \sigma &= 2 \cdot 1 + 2 \cdot \psi, \\
\sigma \times \sigma \times \sigma \times \sigma \times \sigma &= 4 \cdot \sigma,
\end{aligned}
\tag{18}
$$

and so on. The basis in the constrained fusion space can be given by the states

$$
\{|(\sigma\sigma)\sigma; 1\sigma; \sigma\rangle, |(\sigma\sigma)\sigma; \psi\sigma; \sigma\rangle\},
\tag{19}
$$

that correspond to the two left-most $\sigma$ anyons fusing into either 1 or $\psi$. The $F$-matrix to change the basis to fusing from right to left is given by

$$
F = F_{\sigma\sigma\sigma}^{\sigma} = \frac{1}{\sqrt{2}}\begin{pmatrix} 1 & 1 \\ 1 & -1 \end{pmatrix}.
\tag{20}
$$

Since it creates equal weight superpositions, it means that if the fusion outcome is unique in one basis, in the other basis it is completely random. Thus the different fusion orders of Ising anyons correspond to different bases exactly as the basis for a qubit could be chosen along the $Z$-axis or along $X$-axis. Unlike Fibonacci anyons, the fusion space of Ising anyons has a natural tensor product structure. We see from (18) that the dimension of the fusion space doubles for every added $\sigma$ pair and hence for $2N$ anyons the dimension of the fusion space in a fixed topological charge sector is given by $2^{N-1}$.

The $R$-matrix describing the statistics of Ising anyons under the clockwise exchange of the two left-most $\sigma$ anyons is given by

$$
R = \begin{pmatrix} R_{\sigma\sigma}^{1} & 0 \\ 0 & R_{\sigma\sigma}^{\psi} \end{pmatrix} = e^{-i\frac{\pi}{8}}\begin{pmatrix} 1 & 0 \\ 0 & e^{i\frac{\pi}{2}} \end{pmatrix}.
\tag{21}
$$

We immediately see that if we encoded a qubit in the fusion space associated with three $\sigma$ particles, $R^2$ would implement the logical phase-gate up to an overall phase factor. If the two right-most anyons were instead exchanged twice, the evolution in the basis (19) would be described by

$$
F^{-1}R^2F = e^{-i\pi 4}\begin{pmatrix} 0 & 1 \\ 1 & 0 \end{pmatrix}.
\tag{22}
$$

In other words, braiding them changes the fusion channel of the two left most ones between 1 and $\psi$. If the three $\sigma$'s were employed to encode a qubit, this braid would have implemented a logical $NOT$-gate.

The limitation of the operations that can be performed is obvious from the braiding of three $\sigma$ anyons. Since one can only implement logical phase and $NOT$-gates on a single qubit, Ising anyons can only implement the Clifford group by braiding [97, 98]. This means that Ising anyons, while being non-Abelian, are not universal for quantum computation by braiding. To overcome this shortcoming, non-topological schemes have been devised to promote their computational power to universality. While the need for such non-topological operations makes the system more susceptible to errors, Ising anyons are still the best candidates to experimentally test the principles of topological quantum computation due to their realization as Majorana zero modes in superconducting nanowire heterostructures [66–70]. In the Section 4 we describe in more detail how a topological quantum computer would in general be operated. In Section 5 illustrate these steps in the context of the experimentally relevant Majorana wires.

# 4 Quantum computation with anyons

While discussing non-Abelian anyons and the non-local fusion space associated with them, we have already hinted how this space could be used to encode and process quantum information in an inherently fault-resilient manner. In the ideal conditions of zero temperature and infinite anyon separation, the states in the fusion space have three very attractive properties:

(i) All the states are perfectly degenerate.

(ii) They are indistinguishable by local operations.

(iii) They can be coherently evolved by braiding anyons.

If this space of states is used as the logical space of a quantum computer, property (i) implies that the encoded information is free of dynamical dephasing, while property (ii) means that it is also protected against any local perturbations. Property (iii) means that errors could only occur under unlikely non-local perturbations to the Hamiltonian that would create virtual anyons and propagate them around the encoding ones. However, braiding of the anyons can still be carried out robustly by the operator of the computer to execute desired quantum gates. Furthermore, property (iii) implies that all the quantum gates are virtually free from control errors since they depend only on the topological characteristics of the braiding evolutions given by the $F$- and $R$-matrices.

Together these properties mean that quantum computation with anyons would heavily suppress errors already at the level of the hardware, with little need for resource intensive quantum error correction. Of course, in the real world these ideal conditions are never met and some decoherence of the encoded information always takes place. Still, the topological encoding and processing of quantum information provides in principle unparalleled protection over non-topological schemes. For the time being we forget about the nasty reality and focus on outlining the steps to operate a topological quantum computer with Ising anyons as our case study. Generic error sources present in a laboratory are discussed at the end of the section.

## 4.1 Initialization of a topological quantum computer

To initialize a quantum computer, one needs first to identify the computational space of $n$ qubits. In topological computer this means creating some number of anyons from the vacuum and fixing their positions. The system then exhibits a fusion space manifesting as a protected non-local subspace that is identified as the computational space. Depending on the anyons in question, the full fusion space may not admit a tensor product structure (such as the Fibonacci anyons), but one can always identify subspaces corresponding to the fusion channels of subsets of anyons that can serve as qubits. To illustrate the steps of operating a topological quantum computer, we focus on the simpler Ising anyons whose fusion space does exhibit natural tensor product structure.

As discussed in Section 3.4, for $2N$ Ising anyons the fusion space dimension in a fixed topological charge sector increases exponentially as $D = 2^{N-1}$. To initialize the system, we assume that $\sigma$ anyons are created pairwise from the vacuum with no other anyons present. This means that every pair is initialized in the $\sigma \times \sigma \rightarrow 1$ fusion channel and hence the system globally belongs also to the vacuum sector. Thus four $\sigma$ anyons encodes a qubit, six $\sigma$'s encodes two qubits, and so on. Let us focus on a system of six $\sigma$ anyons that enables to demonstrate all the basic operations. It is convenient to choose the pairwise fusion basis as a computational

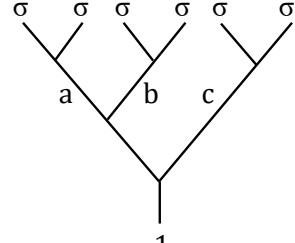

| | **a** | **b** | **c** |
|---|---|---|---|
| $\lvert 00\rangle$ | 1 | 1 | 1 |
| $\lvert 10\rangle$ | $\psi$ | $\psi$ | 1 |
| $\lvert 01\rangle$ | 1 | $\psi$ | $\psi$ |
| $\lvert 11\rangle$ | $\psi$ | 1 | $\psi$ |

Figure 7: The fusion diagram for six Ising anyons for a pairwise basis restricted to the global vacuum sector. Due to the fusion rules $\sigma \times \sigma = 1 + \psi$ and $\sigma \times \psi = \sigma$, the fusion diagram contains four distinct fusion channels that are consistent with the constraint that the fusion of all the six $\sigma$ anyons must give the vacuum 1. The table shows the identification of the fusion channels with the computational basis of two qubits.

basis

$$
\begin{aligned}
\lvert 00\rangle &= \lvert\sigma\sigma;1\rangle\lvert\sigma\sigma;1\rangle\lvert\sigma\sigma;1\rangle, \\
\lvert 10\rangle &= \lvert\sigma\sigma;\psi\rangle\lvert\sigma\sigma;\psi\rangle\lvert\sigma\sigma;1\rangle, \\
\lvert 01\rangle &= \lvert\sigma\sigma;1\rangle\lvert\sigma\sigma;\psi\rangle\lvert\sigma\sigma;\psi\rangle, \\
\lvert 11\rangle &= \lvert\sigma\sigma;\psi\rangle\lvert\sigma\sigma;1\rangle\lvert\sigma\sigma;\psi\rangle,
\end{aligned}
\tag{23}
$$

where the three kets refer to the fusion channels of the three $\sigma$ pairs, as illustrated in Figure 7. When the anyon pairs are created from the vacuum, the topological quantum computer is initialized in the state $\lvert 00\rangle$. The other basis states involve an even number of intermediate $\psi$ channels, which according to the fusion rule $\psi \times \psi = 1$ is consistent with the constraint that the fusion of all $\sigma$ particles must always yield the vacuum.

## 4.2 Quantum gates – Braiding anyons

To perform a computation in the fusion space is equivalent to specifying a braid – a sequence of exchanges of the anyons that corresponds to the desired sequence of logical gates. For Ising anyons, the natural gate set to implement consists of Clifford operations on single qubits, i.e. the $X$-, $Z$- and Hadamard $U_H$-gates and the two-qubit controlled phase gate $U_{CZ}$.

The single qubit gates follow directly from the $F$- and $R$-matrices of Ising anyons, (21) and (22). The first says that under two exchanges the state acquires a $-1$ phase whenever the exchanged $\sigma$ pair fuses to a $\psi$, while the latter says that exchanging twice two $\sigma$ anyons from different pairs simultaneously changes the fusion channel of both pairs between 1 and $\psi$. In the two-qubit computational basis (23), the elementary single qubit operations, up to an overall phase, are thus given by

$$
\begin{aligned}
X_1 &= R_{23}^2 = F^{-1}R^2 F \otimes \mathbb{I}, & Z_1 &= R_{12}^2 = R^2 \otimes \mathbb{I}, \\
X_2 &= R_{45}^2 = \mathbb{I} \otimes F^{-1}R^2 F, & Z_2 &= R_{56}^2 = \mathbb{I} \otimes R^2,
\end{aligned}
\tag{24}
$$

where $R_{ij}$ is the clockwise exchange operator acting on anyons $i$ and $j$. The corresponding braiding diagrams are illustrated in Figure 8. Similarly, the Hadamard gates $U_H$ can be implemented by single exchanges. One can easily verify that $F^{-1}RF$ creates superpositions of fusion channels, but with distinct phase factors assigned to different fusion channels. These phase factors can be cancelled by appending it further exchanges. A little algebra shows that up to an overall phase, the Hadamard gates on the two qubits are given by

$$
\begin{aligned}
U_{H,1} &= R_{12}R_{23}R_{12} = RF^{-1}RFR \otimes \mathbb{I}, \\
U_{H,2} &= R_{56}R_{45}R_{56} = \mathbb{I} \otimes RF^{-1}RFR,
\end{aligned}
\tag{25}
$$

$$X_1 = (R_{23})^2 \qquad\qquad Z_1 = (R_{12})^2$$

$$U_{H,1} = R_{12}R_{23}R_{12} \qquad\qquad U_{CZ} = R_{12}^{-1}R_{56}^{-1}R_{34}$$

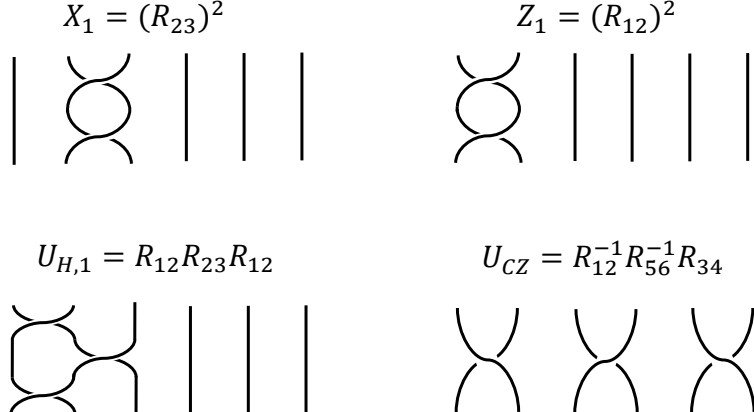

Figure 8: The elementary braids corresponding to the $X$-, $Z$- and Hadamard $U_H$-gates on the first qubit as well as the controlled-$Z$ gate $U_{CZ}$. The single qubit operations on the second qubit are given by (24) and (25).

as illustrated in Figure 8. Hence by braiding one can implement all the single qubit operations in the Clifford group.

To implement two-qubit operations, one needs to perform braids outside this group of operations. The natural two-qubit gate to implement with Ising anyons is the controlled phase gate, that is given by the braid

$$U_{CZ} = R_{12}^{-1}R_{34}R_{56}^{-1}. \tag{26}$$

Since the three exchanges act on separate pairs of $\sigma$ anyons, the action of such braid can be deduced piecewise in the computational basis (23). First, $R_{34}$ gives rise to unitary that maps $|01\rangle, |10\rangle \to e^{i\frac{\pi}{2}}|01\rangle, e^{i\frac{\pi}{2}}|10\rangle$ by assigning the phase factor $e^{i\frac{\pi}{2}}$ whenever the middle pair is in the $\psi$ fusion channel. Likewise, $R_{12}^{-1}$ acts on the left pair and maps $|10\rangle, |11\rangle \to e^{-i\frac{\pi}{2}}|10\rangle, e^{-i\frac{\pi}{2}}|11\rangle$, while $R_{56}^{-1}$ acts on the right pair and maps $|01\rangle, |11\rangle \to e^{-i\frac{\pi}{2}}|01\rangle, e^{-i\frac{\pi}{2}}|11\rangle$. Hence, the joint effect is to map only $|11\rangle \to -|11\rangle$, which is precisely the controlled phase gate between the two qubits.

Unfortunately, this is the best one can do with Ising anyons. A two-qubit gate together with Clifford group operations does not form a universal gate set for quantum computation. For that one requires an additional non-Clifford gate, such as the $\frac{\pi}{8}$-phase gate [99,100]. In principle, it can be implemented non-topologically by bringing two anyons nearby. An interaction between the anyons will then lift the degeneracy of the fusion channels by $\Delta E$ and they will dephase in time according to

$$U = \begin{pmatrix} 1 & 0 \\ 0 & e^{i\Delta E t} \end{pmatrix}. \tag{27}$$

Assuming that bringing the anyons nearby and separating them again can be controlled precisely such that $\Delta E t = \pi/4$, than one would have implemented the desired $\frac{\pi}{8}$-phase gate, albeit in a non-topologically protected fashion. While this is likely to be noisy in the laboratory, i.e. there is likely to be error in the induced phase shift, more elaborate schemes have been proposed to implement this gate in a robust manner [101–103].

## 4.3 Measurements – Fusing anyons

The final step of a computation is the read-out. As the computational basis states (23) correspond to distinct patters of pairwise fusion outcomes, the projective measurements are performed by bringing the anyons pairwise physically together and detecting the fusion outcomes.

How this is done in practice depends on the anyons in question and on the microscopic physics that give rise to them. For the Ising anyons there is in principle a straightforward way to do this. A pairwise fusion can result only either in the vacuum (nothing remains) or a $\psi$ particle (a massive particle remains). This distinction in the change of energy of the system is in principle detectable and serves as a method to perform measurements in the computational basis.

In our example of two qubits encoded in six $\sigma$ anyons illustrated in Figure 7, a projective $Z$-basis measurement of the first qubit corresponds to detecting the fusion outcome of anyons 1 and 2. If no change in energy is detected, then one has applied the projector $|0\rangle\langle0|\otimes\mathbb{I}$, while observing a change in energy applies the projector $|1\rangle\langle1|\otimes\mathbb{I}$. Before the measurement is applied the state of the system could be in a superposition of fusion states. Similarly, an $X$-basis measurements are performed by detecting the fusion outcome of anyons 2 and 3. For measurements on the second qubit the same operations are applied to anyons 5 and 6 for $Z$-basis measurements and to anyons 4 and 5 for $X$-basis measurements. By measuring the fusion outcomes of other anyon pairs, one can apply projectors into joint subspaces of the two qubits. For instance, detecting that anyons 3 and 4 fuse to vacuum projects into the subspace spanned by $|00\rangle$ and $|11\rangle$, while observing $\psi$ as the fusion outcome projects onto its complement.

In a nutshell, these are the basic steps of operating a topological quantum computer. How they are carried out precisely depends on the microscopics of the system that supports the anyons. In Section 5 we apply these ideas to a concrete setting. We outline how a topological quantum computation could in principle be carried in one microscopic system that supports Majorana zero modes, the best realizations we currently have for Ising anyons. Before proceeding there, we briefly review the literature on how a topological quantum computer might fail to deliver the promised protection under realistic conditions.

## 4.4 Possible error sources

The biggest experimental challenge to topological quantum computation is that anyons, especially of the required non-Abelian kind, are hard to come by experimentally. Assuming that one would have access to anyons, there are still several general mechanisms how topological protection might fail despite of its inherent resilience to local perturbations.

**Leakage via spurious anyons**

Like with conventional qubits, decoherence in topological quantum computation can arise due to an uncontrolled coupling to a reservoir. In topologically ordered systems the reservoir could appear in two ways. Either there are additional anyons in the system, causing the full fusion space to enlarge beyond the computational space, or there is another topological system nearby from where quasiparticles can tunnel into and out of the system. Considering that anyons require precise microscopic conditions, it should be relatively easy to take care of the latter by ensuring that there are no accidental topologically ordered states nearby. The first source of a reservoir, however, requires care. As any topological quantum computation is likely to require a macroscopic number of anyons to define a useful computational space (it is estimated that for a robust implementation of quantum algorithms this number ranges from $10^3$ Fibonacci anyons to a whopping $10^9$ Ising anyons [90]), it is in general hard to keep track of all the anyons in the system. When executing quantum gates, braiding around unaccounted anyons would then rotate the state outside the computational space. If the unaccounted anyons are tightly paired and localized, this will not be a problem as their fusion channel is fixed to the vacuum channel and braiding around both has no effect. If the anyons can propagate freely though, accidental braids around isolated anyons can occur leading to decoherence.

**Degeneracy lifting due to anyon-anyon interactions**

The exact degeneracy of the states corresponding to different fusion channels relies on the anyons being non-interacting. In reality, the same microscopics that give rise to them also endow them with interactions that decay exponentially in their separation. Thus at best one can demand them to be far away, enough for interactions become negligible and thus unable to distinguish the fusion channels by lifting their degeneracy. In the laboratory everything is finite though and when a macroscopic number of anyons is required to operate a topological quantum computer, it is impossible to keep all of them well separated at all times. What far away means for a certain system is given by the coherence length that scales in general as the inverse energy gap $\xi \sim \Delta^{-1}$. As the anyons are quasiparticles, that are collective states of the elementary excitations, they are in practice always exponentially localized with the decay of the wave function controlled by $\xi$. Thus when two anyons with two possible fusion channels are within distance $L$ of each other, virtual tunneling processes between them lift the degeneracy of the fusion channels by $\Delta E \sim e^{-L/\xi}$ [104]. Indeed, this splitting has been calculated in $p$-wave superconductors [105], fractional quantum Hall states [106] and spin liquids [107], and also experimentally observed for Majorana zero modes in superconducting nanowires [70].

Lifting of the fusion channel degeneracy implies several subtleties for topological quantum computers. First, logical states with different energies will dephase with time. While they remain insensitive to local operations, the Hamiltonian of the system will distinguish between them akin to the non-topological implementation of the $\frac{\pi}{8}$-phase gate (27). Thus topological qubits, like non-topological ones, can also decohere with time unless error correction is applied [108]. Second, quantum gates by braiding are no longer necessarily exact. Finite energy splitting between the ground states implies that the evolution must be fast enough at the scale of $\Delta E$ for the states to appear degenerate, while still being slow enough at the scale of the energy gap $\Delta$ not to excite the system [109, 110]. In general this balancing between the two energy scales means that there are small errors in the implemented logical operations. If they accumulate and are not error corrected, they will become a source of decoherence.

Anyon-anyon interactions can also induce topological phase transitions when the anyons form regular arrays [111–113]. Any scalable architecture for a topological quantum computer is likely to employ a systematic arrangement of anyons to keep track of them. As a transition to a different topological phase of matter is the ultimate failure of a topological quantum computer, it needs to be avoided at all costs. The microscopic conditions for such anyon-anyon interaction induced phase transitions, particularly those for arrays of localized Majorana zero modes, have been studied in several works [114–119].

**Finite temperature**

Finite temperature means that one no longer considers pure states of fixed number of excitations, but mixtures of states corresponding to different number of excitations. In principle the energy gap $\Delta$ of topologically ordered systems protects the encoded information by suppressing such thermal fluctuations at least as $e^{-\frac{\Delta}{kT}}$. However, even very small weights of thermally excited states can cause a problem in intrinsic topological states where the excitations are anyons. Studies on topological quantum memories based on Abelian anyons suggest that any finite temperature, even if much smaller than the gap, can cause the encoded information to be lost and this is expected to be an issue also for non-Abelian anyons [120–122]. Any finite system, and in a laboratory everything is finite, does have a critical threshold temperature. However, as an extensive amount of anyons is in general needed, such protection in realistic finite-size systems is unlikely to be sufficient for stability. Promisingly, this may be circumvented if the system possesses some mechanism that suppresses the spontaneous creation of

stray anyons [9, 123–126], but it is unclear whether such schemes are realistic. Experimentally the situation is equally challenging as the energy gaps tend to be small and thus only formidably low temperatures can be tolerated even for short times.

Regarding the problem of finite temperature, SPT states that require defects to bind anyons might actually be more stable than intrinsic topological states, since not all defects are thermally excitable (e.g. a small temperature can not cut wires into pieces or create lattice dislocations). We defer the discussion of thermal stability in the Majorana zero mode hosting superconducting nanowires to Section 5.4.

# 5   Topological quantum computation with superconducting nanowires

We turn now to illustrating the steps of topological quantum computation, as outlined in the previous section, in the context of a microscopic model. As an example we consider Kitaev's toy model for a superconducting nanowire [63]. This model gives rise to a 1D superconducting SPT state where Majorana zero modes are bound at the ends of the wire or at at domain walls between topological and trivial phases. The Majorana zero modes are for all practical purposes equivalent to Ising anyons. The technical difference is that the braiding of Majorana modes bound to defects realizes the $R$-matrices of Ising anyons only projectively [149], i.e. the overall phase factors in (21) and (22) are omitted. However, since overall phases are not relevant to quantum computation, quantum computating with Majorana zero modes proceeds exactly as outlined in Section 4.

Long before Majorana modes acquired experimental relevance, it was known that quasiparticles in a two-dimensional system described by localized Majorana modes would exhibit the braiding statistics of Ising anyons [28, 54]. Formally, a Majorana mode is "half" of a complex fermion mode. By this we mean that if $f$ is a fermion operator satisfying $\{f^\dagger, f\} = 1$, one can always write

$$f = \frac{1}{2}(\gamma_1 + i\gamma_2), \tag{28}$$

where $\gamma_i = \gamma_i^\dagger$ are Hermitian Majorana operators satisfying $\{\gamma_i, \gamma_j\} = 2\delta_{ij}$ and $\gamma_i^2 = 1$. If two Majorana modes, $\gamma_1$ and $\gamma_2$, would exist as quasiparticles localised at different positions, then the occupation $f^\dagger f = 0, 1$ of the complex fermion shared by them would constitute a non-local degree of freedom. This would precisely correspond to the non-local degree of freedom of two $\sigma$ anyons described in Section 18. If the fermionic mode is unoccupied ($f^\dagger f = 0$), then the two Majorana modes would behave like the vacuum when brought together, while if it is occupied ($f^\dagger f = 1$), then the fusion of two such quasiparticles would leave behind the fermion $\psi$.

Writing a complex fermion operator as a linear combination of two Majorana operators is a mathematical identity, that by no means implies that Majorana modes could actually appear on their own. They are named after Ettore Majorana, who in 1937 suggested that 3D fermionic particles that are their own antiparticles could describe neutrinos. Their restriction to 2D brings about a non-Abelian anyonic behaviour to the wave function that describes them, which is not present in their 3D counterpart. Such realizations can be met in condensed matter systems. Pioneering work on $p$-wave superconductors [54] suggested though that such exotic 2D superconductors might host Majorana modes localized at vortices. Building on this work, Kitaev proposed a simplified 1D toy model where Majorana modes could appear at the ends of a superconducting wire [63]. While not having a clear experimental realization at the time, this model provided the invaluable insight that domain walls can also host Majorana modes. It took 10 years to discover that Kitaev's simple model could be physically realized by deposit-

ing a spin-orbit coupled semiconducting nanowire on top of a normal $s$-wave superconductor and subjecting it to suitably oriented magnetic fields [64, 65]. Remarkably, a few years later experimentalists confirmed this prediction [66–69, 71]. As discussed in Section 2, numerous microscopically distinct proposals for realizing Majorana modes have since been put forward. Regardless of the microscopic description, the low energy physics can always be cast in the form of the original toy model [63].

## 5.1 Majorana zero modes in a superconducting nanowire

Consider a system of spinless superconducting fermions on a 1D lattice of length $L$ described by the Hamiltonian

$$H = \sum_{j=1}^{L} \left[ -t \left( f_j^\dagger f_{j+1} + f_{j+1}^\dagger f_j \right) - \mu \left( f_j^\dagger f_j - \frac{1}{2} \right) + \left( \Delta_p f_j f_{j+1} + \Delta_p^* f_{j+1}^\dagger f_j^\dagger \right) \right]. \qquad (29)$$

Here $t$ is the tunnelling amplitude, $\mu$ is the chemical potential and $\Delta_p = |\Delta_p| e^{i\theta}$ is the superconducting pairing potential. Following [63], this Hamiltonian for $L$ complex fermions can be written in terms of $2L$ Majorana operators. Absorbing the superconducting phase $\theta$ into the definition (28) by writing $f_j = e^{-i\theta/2}(\gamma_{2j-1} + i\gamma_{2j})/2$, the Hamiltonian takes the form

$$H = \frac{i}{2} \sum_{j=1}^{L} \left[ -\mu \gamma_{2j-1} \gamma_{2j} + (t + |\Delta_p|) \gamma_{2j} \gamma_{2j+1} + (-t + |\Delta_p|) \gamma_{2j-1} \gamma_{2j+2} \right], \qquad (30)$$

which, as illustrated in Figure 9(a), now describes free Majorana operators on a 1D lattice of length $2L$ subjected to nearest and third-nearest neighbour tunneling.

There are two limiting coupling regimes where the ground state of $H$ can be obtained immediately. When the chemical potential term dominates, we can set $|\Delta_p| = t = 0$. The Hamiltonian becomes

$$H = -\frac{i}{2} \sum_{j=1}^{L} \mu \gamma_{2j-1} \gamma_{2j} = -\mu \sum_{j=1}^{L} (f_j^\dagger f_j - \frac{1}{2}), \qquad \mu \gg t, |\Delta_p|, \qquad (31)$$

which means that the ground state is given by having a fermion ($f_j^\dagger f_j = 1$) on every site, as illustrated in Figure 9(b). This state is a product state of fermion modes localized on physical sites and hence topologically trivial.

The other limiting coupling regime is to have the kinetic term comparable to the pairing potential and dominating over the chemical potential. Setting $t = |\Delta_p|$ and $\mu = 0$, we obtain the Hamiltonian

$$H = it \sum_{j=1}^{L} \gamma_{2j} \gamma_{2j+1} = 2t \sum_{j=1}^{L-1} (\tilde{f}_j^\dagger \tilde{f}_j - \frac{1}{2}), \qquad t = |\Delta_p| \gg \mu, \qquad (32)$$

where we have defined a new set of fermionic operators by combining the Majoranas as $\tilde{f}_j = e^{-i\theta/2}(\gamma_{2j} + i\gamma_{2j+1})/2$ from different physical sites. As illustrated in Figure 9(b), the Majorana operators $\gamma_1$ and $\gamma_{2L}$ at the edges completely decouple from the Hamiltonian that now describes interactions only between $L-1$ complex fermions. The missing fermion can be described by the operator $d = e^{-i\theta/2}(\gamma_1 + i\gamma_{2L})/2$ that is delocalised between the two ends of the wire. Since $[d^\dagger d, H] = 0$, the ground state in this limit is two-fold degenerate. Both degenerate states have no localized fermions in the bulk of the system ($\tilde{f}_j^\dagger \tilde{f}_j = 0$), but they differ by the population of the delocalized fermion mode ($d^\dagger d = 0, 1$). Formally, this mode can be added to Hamiltonian (32) by assigning zero energy to it. For this reason the edge

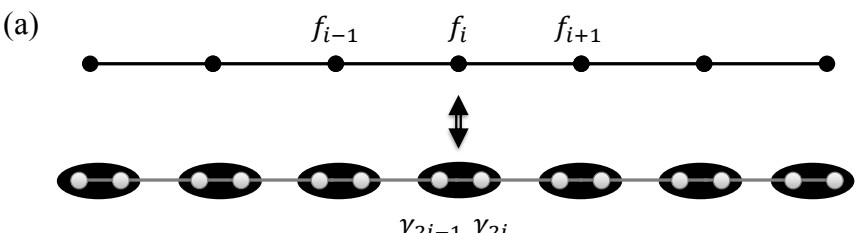

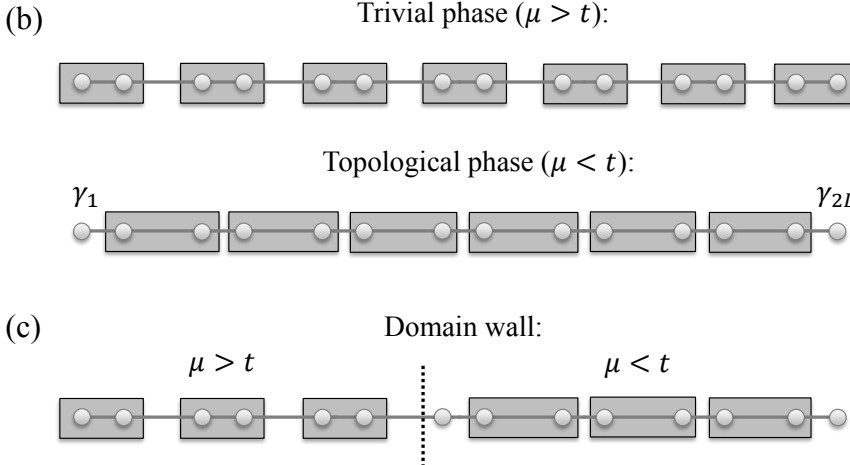

Figure 9: Kitaev's toy model for a $p$-wave paired superconducting wire [63]. (a) The original Hamiltonian (29) in terms of complex fermions $f_j$ is defined on a one-dimensional lattice of $L$ sites. When each complex fermion operator is decomposed into two Majoranas by writing $f_j = e^{-i\theta/2}(\gamma_{2j-1} + i\gamma_{2j})/2$, the Hamiltonian (30) describes free Majorana fermions on a chain of length $2L$. (b) When $\mu \gg t, |\Delta_p|$, the system is in the trivial phase described by the Hamiltonian (31), whose unique ground state has all the Majorana modes paired. In the opposing limit, the system is in the topological phase where the Majorana operators $\gamma_1$ and $\gamma_{2L}$ strongly decouple from the Hamiltonian (32). The localized end states constitute a single non-local zero energy fermion mode $d = (\gamma_1 + i\gamma_{2L})/2$ and the ground state is thus two-fold degenerate. (c) If a part of the wire is in topological phase and the other part in the trivial phase, then the domain wall separating the phases also binds a localized Majorana mode.

Majorana modes appearing in topological wires are also called Majorana zero modes. As opposed to the trivial state in the $t = |\Delta_p| = 0$ limit, we call this state topological since it exhibits the defining characteristic of an SPT state: For open boundary conditions the ground state exhibits degeneracy that arises from edge states (here the two Majorana zero modes due to the delocalized fermion mode), while for periodic boundary conditions the ground state is unique (the fermion mode $d$ is no longer delocalized and has finite energy). More formally, this state is also characterised by a non-trivial topological invariant [63].

The coupling regimes $t = |\Delta_p| = 0$ or $\mu = 0$ are limiting cases that demonstrate that the Hamiltonian (29) for the spinless superdonducting wire supports two distinct phases of matter, one of which is topological with Majorana modes appearing on the wire ends. These wire ends should be viewed as defects, that bind the anyons (even if the system is in the topological phase, for periodic boundary conditions there are no localized Majorana modes). Of course, these idealized limits are never realized in a realistic setting, but that is not necessary either. As two topologically distinct gapped states, they must both be part of extended phases in the

parameter space. The Hamiltonian can be solved exactly for all values of the parameters via
Fourier transform followed by a Bogoliubov transformation [63], which gives a phase diagram
where the topological and trivial phase are separated by a phase transition at $t = |\mu|$ (we set
here $t = |\Delta_p|$). The topological phase with Majorana modes at the wire ends thus persists as
long as $t > |\mu|$.

The consequence of being away from the idealized $\mu = 0$ point is that the Majorana modes
will be exponentially localised at the wire ends rather than being positioned on a single site.
The operators describing them at the left (L) and right (R) ends of the wire take the general
form

$$\Gamma_{L/R} = \sum_{j=1}^{2L} \alpha_j^{L/R} \gamma_j. \tag{33}$$

The normalised amplitudes decay as $|\alpha_j^L| \propto e^{-j/\xi}$ and $|\alpha_j^R| \propto e^{-(2L-j)/\xi}$, where $\xi \propto \Delta^{-1}$ is
the coherence length and $\Delta$ the spectral gap, that depends on the precise values of $t$ and $\mu$,
that separates the degenerate ground states from the rest of the spectrum. This means that
for finite wires of length $L$, the wave functions of the two Majoranas will in general overlap,
which in turn results in a finite energy splitting $\Delta E \propto e^{-L/\xi}$ between the two ground states.
This splitting is negligible for long wires. Nevertheless, it serves as an explicit reminder that
topologically ordered phases emerge always in finite microscopic systems. The anyon model
provides an exact effective low energy description of the system only in an idealised limit
of infinite system size and energy gap. The anyons (the Majorana modes at the wire ends)
are collective quasiparticle states of underlying more fundamental particles (electrons in the
wire), that have also microscopic dynamics of their own. Indeed, such degeneracy lifting due
to finite wave function overlap, that can be viewed as an anyon-anyon interaction as discussed
earlier, has been predicted in microscopically distinct nanowires [145–148] and subsequently
experimentally observed [70].

## 5.2   The Majorana qubit

Recall that in Section 3.4 we discussed the Ising anyon model that consists of three particles 1,
$\psi$ and $\sigma$ obeying the fusion rules (18). If the two localised Majorana modes $\gamma_1$ and $\gamma_{2L}$ at the
ends of the wire can be viewed as a pair of $\sigma$ anyons, how should one view the vacuum 1 and
the fermion $\psi$? Since 1 denotes trivial topological charge, it is clear that it should be identified
with the ground state of the topological phase in the absence of any excitations or Majorana
zero modes. The intrinsic excitations of the system are fermionic quasiparticles, created by the
operators $\tilde{f}_j^\dagger$ appearing in the diagonalized spectrum (32). These should be identified with the
$\psi$ particles of the Ising anyon model, which gives a natural interpretation to the fusion rule
$\psi \times \psi = 1$. As with any superconducting system, the ground state is a condensate of Cooper
pairs of fermions and the elementary excitations are the fermions obtained by breaking the
pairs up at the energy cost of $2\Delta$, i.e. twice the energy gap due to $\psi$ particles always coming
in pairs. The fusion rule just states that two fermions can pair up to form a Cooper pair that
vanishes into the condensate, thus becoming part of the ground state 1. This also means that
only the parity of the $\psi$ fermions is conserved. Indeed, the only exact global symmetry of the
Hamiltonian (29) is fermion parity described by the operator $\mathscr{P} = \exp(i\pi \sum_j f_j^\dagger f_j)$.

With these identifications in mind, we now assume that the system contains only Majorana
zero modes corresponding to $\sigma$ anyons, but no fermions $\psi$. When only two Majorana modes
are present, the occupation of the non-local fermion shared by them is described by the opera-
tor $d^\dagger d = (1 + i\gamma_1 \gamma_{2L})/2$. The state with eigenvalue of $d^\dagger d$ being 0 can then be identified with
the fusion channel $\sigma \times \sigma \to 1$, while the state with eigenvalue 1 corresponds to $\sigma \times \sigma \to \psi$.
As a result, the two degenerate ground states in the topological phase can be identified with

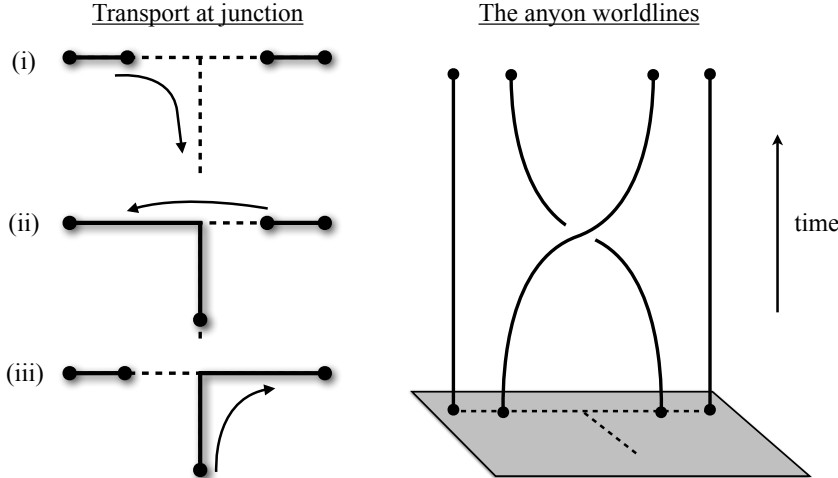

Figure 10: Braiding Majorana modes at a T-junction of superconducting nanowires, where the chemical potential $\mu$ in the Hamiltonian (29) is locally tunable. Dashed lines denote domains in the trivial phase ($|\mu| > t$), solid lines domains in the topological phase ($|\mu| < t$) and the dots denote the locations of Majorana zero modes. In the even fermion parity sector ($\mathscr{P} = 1$) the two-fold degenerate ground state of this system is spanned by the states (35) and separated from excited states containing an even number $\psi$ excitations by an energy gap of $2\Delta$. By locally tuning the chemical potential $\mu$ along the wires, one can extend or contract the topological domains and thereby move the domain walls binding the Majorana modes. The shown sequence (i)-(iii) results in the Majorana modes of different topological domains to be exchanged such that their worldlines are braided. When done adiabatically, the non-Abelian Berry phase (5) for this exhange evaluates to the braid matrix $R_{23}$ of the Ising anyon model, as discussed in Section 4.3.

the two fusion channel states

$$i\gamma_1\gamma_{2L}|\sigma\sigma;1\rangle = -|\sigma\sigma;1\rangle, \qquad i\gamma_1\gamma_{2L}|\sigma\sigma;\psi\rangle = +|\sigma\sigma;\psi\rangle, \tag{34}$$

where $|\sigma\sigma;\psi\rangle = d^\dagger|\sigma\sigma;1\rangle$. However, as we already discussed in Section 4.1, these two states cannot form a basis for a qubit, because they belong to different $\psi$-parity sectors. In the context of the superconducting wire this is seen be studying the action of the fermion parity operator $\mathscr{P}$ on the two states (34). On them it acts as $\mathscr{P} = i\gamma_1\gamma_{2L}$, which means they belong to different parity sectors and thus one can not form coherent superpositions of them.

To form a qubit one needs two wires hosting four Majorana zero modes [150–152]. Such system has a ground state manifold that contains two states belonging to each parity sector. Choosing the even parity sector ($\mathscr{P} = 1$), the computational basis states can be identified with the fusion channel states (19) as

$$|0\rangle \equiv |\sigma\sigma;1\rangle|\sigma\sigma;1\rangle, \qquad |1\rangle \equiv |\sigma\sigma;\psi\rangle|\sigma\sigma;\psi\rangle, \tag{35}$$

where the two kets now refer to the fusion channel states of the two wires. One can thus view every wire with Majorana end states as a pair of $\sigma$ anyons created from the vacuum. By adding a third wire, the computational basis of two qubits could be defined precisely as (23), as discussed in Section 4.1.

## 5.3   Manipulating and reading out the Majorana qubit

The fact that two distinct Majorana wires are required to encode a qubit might seem like posing a problem for manipulating it via braiding. As braiding involves moving the anyons around each other, the obvious problem is that the Majorana modes are stuck at the ends of disjoint wires whose position is fixed. However, a wire end is nothing but a domain wall between the topological phase and the vacuum, which is equivalent to the trivial phase. Thus Majorana modes exist also at the domain walls that separate the topological phase (a region where $t > |\mu|$) from the trivial phase (a region where $t < |\mu|$), in the same physical system, as illustrated in Figure 9(c). By locally controlling, say, the chemical potential $\mu$ by gating the wire, even a single physical wire can realize many sections that each contribute a single Majorana wire. By adiabatically changing the chemical potential locally, the domain walls, and thus the Majorana modes bound to them, can in principle be moved along the wires.

To perform braiding, consider a T-junction of these wires, as illustrated in Figure 10. The junction structure enables exchanging Majorana modes both from same and different topological domains and thereby implement via braiding the single qubit Clifford operations (24) discussed in Section 4.2. While the explicit calculation is beyond the scope of this introduction, Alicea et al. [133] have shown that when such exchanges are performed by adiabatically tuning the chemical potential along the wires, the resulting non-Abelian Berry phase (5) in the two-dimensional ground state manifold spanned by the states (35) indeed coincides, up to an overall phase, with the braid matrices of the Ising anyon model. The $\frac{\pi}{8}$-phase gate, that is required for universality, can similarly be implemented by locally tuning the chemical potential to bring two domain walls nearby and letting them dephase for a precise time according to (27). To promote the T-junction into a topological quantum computer, more Majorana qubits can be realized and manipulated in extended arrays of junctions where many different sections are either in the trivial or in the topological phase. The realization of such wire arrays is a challenge on itself, but promisingly there is both theoretical [153,154] and even experimental evidence [155] for their feasibility.

Moving anyons around by manipulating the chemical potential is a direct, but not the only method to implement braiding evolutions. In fact, since tuning the chemical potential involves quenching the system locally into and out of the topological phase to move the domain walls, it can result in unwanted excitations that are sources of error [156,157]. To circumvent this, several more desirable schemes have been devised to implement effective braiding evolutions that do not require actually moving the anyons around. One promising scheme involves switching domains of the wires between Josephson energy and charging energy dominated regimes enabling Majorana modes to jump between different domain walls without the need of precise local control [158–161]. A blueprint for implementing a topological quantum computer based on such ideas can be found in [162]. An alternative is the so called measurement-only topological quantum computation, where non-destructive measurements of anyon charges have been shown to simulate arbitrary braid evolutions [163, 164]. A blueprint to implement this scheme with Majorana zero modes in superconducting wires has been put forward in [165].

Assuming that the braiding evolutions can be implemented by some means, the final step of a topological quantum computation is the read out, as discussed in Section 4.3. As can be seen from the basis states (35), to measure the Majorana qubit in the $Z$-basis, one needs to measure the fermionic population of either of the two Majorana pairs constituting the qubit. This can be performed by adiabatically bringing the Majorana modes close to each other by slowly shrinking the topological domain they belong to. When they are superposed, fusion takes place and the state of the system becomes locally either the vacuum 1 or the fermion $\psi$ depending on the occupation $d^\dagger d$ of the non-local fermion mode. Since $\psi$ is a massive excitation, there is a relative energy difference of $\Delta$ between these two alternatives. Detecting this shift in the energy of the system after fusing the Majorana modes amounts to performing a projective

measurement on the qubit. Likewise, as also discussed in Section 4.3, a measurement in the $X$-basis can be similarly implemented by fusing Majorana modes from different topological domains and detecting the shift in the energy of the system. For details on how these changes in energy can be detected in the microscopic architecture, we refer the interested reader to [162, 165].

## 5.4  Challenges with Majorana-based topological quantum computation

In Section 4.4 we discussed general challenges that can affect topological quantum computation. Regarding the presence of unwanted anyons, these can appear in Majorana wires due to disorder. Like domain walls can be moved by tuning chemical potential, disorder along the wire, exhibited by a locally random chemical potential, can cause parts of the wire to be accidentally in either the topological or the trivial phase. This causes additional domain walls along the wire that will host additional unaccounted Majorana modes [166, 167]. These can cause leakage out of the computational space, but encouragingly this effect can be mitigated when employing Josephson-charging energy switching braiding protocols [168].

Majorana-based schemes can also suffer from the leakage to an external reservoir. The Majorana qubit is protected by the fermion parity that is exact in a closed system [169]. However, the physical schemes to realize Majorana wires require depositing the spin-orbit coupled nanowires on top of an $s$-wave superconductor [59, 64, 65] from which Cooper pairs tunnel into the wire inducing superconductivity. If Cooper pairs can tunnel from the wire, so can Bogoliubov quasiparticles. In other words, the $s$-wave superconductor can also serve as a reservoir of $\psi$ particles breaking the parity conservation. This sets limits on time-scales how fast the quantum computation needs to be performed before such poisoning occurs [170–173]. Encouragingly, heterostructures for topological nanowires exhibit long poisoning times that should make such errors manageable [174, 175].

Regarding the question of finite temperature, the engineered $p$-wave superconductors may be more robust than intrinsically topologically ordered states. Since Majorana modes are bound to domain walls, they are not spontaneously created by thermal fluctuations. The energy gap protects the localized states on the domain walls from extended states, which suggests that protection by the energy gap should be sufficient. Indeed, studies show that Majorana qubits do tolerate finite temperatures [176, 177], which should thus not pose a fundamental obstacle.

# 6  Outlook

We have seen that the key ingredients for performing topological quantum computation are: (i) To have access to a system supporting non-Abelian anyons, (ii) to be able to adiabatically move them around each other or simulate such evolutions in a topologically protected manner and (iii) to be able measure their fusion channels. If one were to have access to Fibonacci anyons, then these steps would be sufficient to implement universal quantum computation. Unfortunately, the theoretically proposed states that support these particular types of anyons are very fragile and thus experimentally challenging. Thus the research has focused on simpler types of anyons known as Ising anyons. Their free-fermion counterparts – the Majorana zero modes – are strongly believed to exist, with experimental support in heterostructures of spin-orbit coupled nanowires and normal $s$-wave superconductors [66–69, 71]. While not universal for quantum computation by purely topological means, they serve as a test bed for the key selling points of topological quantum computation – non-local encoding, protection by energy gap and quantum gates by braiding. If additional non-topological operations are used, then Majorana fermions are able to implement arbitrary quantum gates. Considering that the

existence of clear hard energy gaps [178] and the exponential localization of the Majorana end states [70] has already been experimentally verified, there is much hope that the proposed experiments on the braiding properties [133, 134, 162] will also be carried out successfully in these systems.

While being very encouraging, it should be kept in mind that Majorana modes do not provide the full power of topologically protected quantum computation nor is any anyon based scheme a panacea for all the troubles of quantum computation. The hardware level protection they provide is highly desirable, but they all come with their own shortcomings to overcome. Still, considering the open problems faced by non-topological schemes and the rapid progress in preparing and controlling topological states of matter hosting Majorana modes, overcoming these challenges is a fair price to pay for the robustness that comes with topological quantum computation. Ways to go beyond Majorana modes already exist based on the same constructions. Replacing the spin-orbit coupled semiconductor nanowires with edge states of Abelian fractional quantum Hall states can realize parafermion modes that allow for a larger, although still non-universal gate set [179,180]. While the Read-Rezayi fractional quantum Hall state [37] hosting Fibonacci anyons might just be too fragile, it has been proposed that collective states of parafermion modes can give rise to an analogous state that supports the Fibonacci anyons [96]. Since Abelian fractional quantum Hall states are well established experimentally, it is not too far fetched to imagine that the current technology can be pushed to realize parafermion modes and, stretching the imagination a bit further, perhaps even Fibonacci anyons. Another exotic idea is to employ superconducting nanowires to engineer intrinsic topological order that supports Ising anyons and employ special defects to promote the Ising anyons for universal quantum computation by topological means [181].

A more realistic route might be some midway that enjoys some of the benefits of topological quantum information storing and processing, while not being fully a topological quantum computer as described here. One such scheme is to construct surface codes that support Abelian anyons from Majorana wire based qubits and design fault-tolerant protocols to promote such systems into universal quantum computers [182]. Another is to couple Majorana qubits to non-topological spin qubits, which enables to realize a universal gate set [183]. Whatever the route taken, the general principles underlying any topological scheme will be based on the basic operations described in this review.

### Funding information

JKP would like to acknowledge the support of EPSRC through the grant EP/I038683/1. VTL is supported by the Dahlem Research School POINT fellowship program.

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
