# Peer review of "A Short Introduction to Topological Quantum Computation"

_SciPost Physics, doi:SciPost Phys. 3, 021 (2017)_

## Round 1 · Referee Report · Anonymous (Referee 1) · 2017-6-2

Strengths

  • Readability and style
  • Length

Weaknesses

  • Unclear intended audience and prerequisites (for an introductory text)
  • Unclear placement with respect to existing literature and online material
  • In several points, lack of (short) explanations / definitions of several advanced concepts

Report

This work is an introduction on topological quantum computation. There are already several reviews/monographs/notes/lectures available on this topic, with different degrees of length, depth, prerequisites. The table of contents and lenght of this work may make it an appropriate early entry point to the subject, and in many parts the text is of good quality. However, it is not clear which type of reader the Authors had in mind when writing the text. This work often seems to assume advanced knowledge of condensed matter physics, and the topological aspects of it in particular, but (of course) no knowledge in topological quantum computation. This is a rather peculiar combination of background knowledge and may make this work not as useful as it could be to the truly uninitiated reader (the initiated readers already have plenty of material and references to increase their knowledge, and this work would likely not be used for that purpose).

Many concepts are introduced without proper explanation, and many references or remarks are made which cannot be useful to someone without an advanced level of research experience in the topic. Sometimes distinct concepts are conflated or thrown around in the same paragraphs, in a way which is easily interpretable by the experts but which may be confusing to beginners. I will point out several occurrences of these problematic passages in the list which follows below.

I believe that re-focusing several passages this work can find its place as an accessible, high-quality entry point to the field. Without such revision, it would remain a text without a clear purpose and without a clear audience, so I hope the Authors will consider this advice. Altogether, I believe SciPost is the right venue for this work.

The list of "requested changes" that follows contains some more open-ended remarks, and some more detailed ones.

Requested changes

This is a list of points where I believe the exposition is problematic. I do not always request an explicit change, but I believe that all these points should somehow be addressed in a revisited exposition.

1 - In the introduction, Sec. 1, the Authors manage to avoid mentioning two fundamental concepts behind the idea of topological quantum computation, adiabaticity and degeneracy. While these are discussed later, I think the two concepts are so important (while also being understandable to anyone with a good basic knowledge of quantum mechanics) that it would be beneficial to mention their crucial role topological quantum computation.

2 - Page 2, "(at least what comes to our every day energy scales)". This is a rather obscure remark, not sure what is meant by that.

3 - The paragraph below Eq. 3 is also quite obscure for the uninitiated reader. First, the concept of fractionalization is introduced here first without any explanation. It is linked immediately to strong interactions, but the connection is not clear. This is an example of a paragraph which would be unintelligible for someone without the proper condensed matter background, for instance a knowledge of fractional quantum Hall effect. Also, note Majoranas do not require strong interactions, yet they are the most treated example in the rest of the work; they are also not so easily linked to fractionalization or the Aharonov-Bohm effect. So it is also not clear that the concept quickly introduced here are beneficial to the reader to understand the material that follows.

4 - In the paragraph below Eq. 3, the sentence "This happens if the charge or the flux is fractionalised, i.e. $q \neq= n$ (in units of the electron charge) or $\Phi\neq 2\pi n$, respectively" is unclear. First, Eq. 3 is only a joint condition on the product $\Phi q$. Also, physical units for flux are off, or a definition of the flux quantum is missing.

5 - The first two paragraphs of Sec. 1.3 are too dense and can not be followed without knowing several advanced concepts of condensed matter physics. At this point in the manuscript a definition or description of topological order has not been attempted yet (not even a heuristic one), yet these paragraphs dwell on the distinction between SPTs and intrinsic topological order. Again, it's not clear what the benefit is for the reader. The role of topology in condensed matter is very broad and diverse and it is not always directly relevant for TQC, especially in a supposedly gentle introduction such as this one.

6 - Last sentence of page 5, "topological insulator heterostructures". This term seem to exclude semiconductor heterostructures, which are actually an even most prominent candidate to engineer topological superconductivity. I would suggest to reformulate the wording to make the term more encompassing of all the different proposals.

7 - Page 6, first paragraph: "off-diagonal conductance" is a confusing and unusual term for the familiar Hall conductance.

8 - Page 6, "Employing the intricate connection to conformal field theory,...". Which connection? No mention of such connection was made so far in the text. This sentence comes out of the blue and, again, is very obscure for someone not knowledgeable about the topic.

9 - Page 7, "... it has been known that if the pairing in a 2D superconductor is so-called p-wave type, i.e. time reversal is broken by the pairing term, then..." Breaking TRS is a necessary but not sufficient condition to obtain p-wave pairing, hence this sentence is slightly misleading. This passage should be expanded not to make things sound more simple than they are.

10 - Page 7, last paragraph before Sec. II. It is not clear from this paragraph if these quantum simulations are only equivalent to topologically ordered states via a non-local mapping, or if they realize the ordered state directly. Judging from the list of references, it probably conflates the two things, which conceptually is not helpful.

11 - In Sec. 2, it would be nice to give some more information about pentagon and hexagon equations. These are mentioned several times but never really discussed, not even at a heuristic level, so a sense of mystr is left in the reader. As a consequence, some passage are obscure. For instance, why do the hexagon equation have more than one solution, and to the pentagon equation also share this property?

12 - Sec. 2.4: "Unlike Fibonacci anyons, the fusion space of Ising anyons has a natural tensor product structure.". This sentence is crucial but also mysterious, because the strucutre of the Hilbert space of Fibonacci anyons was not discussed in the section regarding Fibonacci anyons. It should be explained how the Hilbert space of Fibonacci anyons has no tensor product structure, and why this is special. The same problem features in a similar sentence in the first paragraph of Sec. 3.1. The structure of the fusion space of Fibonacci anyons is invoked twice without prior discussion of the notion.

13 - Page 13, second paragraph, "means that it is also protected against any uncorrelated local perturbations." What does uncorrelated mean here? Is it a redundant adjective or is it essential? Do "correlated" local operations do something in the fusion space?

14 - After Eq. (21), "While this is likely to be challenging in the laboratory..." I would say it is more likely to be very noisy, than challenging. Such a coupling can be easily achieved at least in the case of Coulomb-coupled Majorana modes.

15 - Page 18, "In the presence of thermally excited anyons the statistics of anyons as is expected to become ill-defined and..." First, this sentence is grammatically broken. Second, it is not clear why the statistics of anyons becomes ill-defined.

16 - Sec. IV, first paragraphs. The use of the adjective "massive" is counterproductive here. First, this word has many heuristic meanings depending on the context. Second, I think there is no easy sense in which Majorana zero modes are "massive". What is missing here is a thorough discussion of the distinction between anyons as true quasiparticles (as in the FQHE) or as quasiaparticles associated with topological defects. So again, without further elaborations (which may go beyond the scope of the paper) the use of the term is confusing.

17 - The first pargraph of Sec.4.2 needs serious improvement: first, "the degeneracy of the ground state in the presence of anyonic quasiparticle excitations" is an oxymoron (ground state in presence of excitations); second, anyons may not arise as true excitations of the system (topological superconductors have no anyonic excitations in the strict sense of the term); third, ground state property and properties of the excitations are mixed throughout the paragraph.

18 - "The degeneracy equals the number of distinct anyons (including the vacuum)." This statement is given without a heuristic explanation, and as such it is not very helpful and becomes quite forgettable.

19 - "The degeneracy is most conveniently obtained from the Verlinde formula [124]." This is another concept/formula which is name-dropped without explanation or clear benefit to the reader.

20 - Entanglement entropy is also never defined, not even heuristically. It is quite an advanced concept to assume a beginner reader to be familiar with. Same problem with "quantum dimension" and "topological twists" later in this section.

21 - Page 21, "Since the work of Ivanov, ..." While of course Ivanov's paper is a seminal one, I think the braiding properties of vortices in p+ip superconductor where also discussed in Read & Green.

22 - Page 22, "...implies that Majorana fermions could...". I feel that by now Majorana fermion is a discouraged term in this context. Also, terminology should be consistent and it seems in the rest of the work authors prefer the more precise "Majorana zero modes".

23 - Page 22, "It took 10 years to discover that Kitaev’s simple model could be realized by depositing a spin-orbit coupled semiconducting nanowire on top of a normal s-wave superconductor [56, 57]." I would mention the necessary presence of a magnetic field.

24 - Page 25, "These include switching domains of the wires between Coulomb energy and charging energy dominated regimes enabling Majorana modes to jump between different domain walls [137, 138]". This sentence is wrong, since charging and Coulomb energy dominated regimes are the same thing. The Authors probably meant charging vs Josephson regimes. Also, the References are wrongly placed here; the Coulomb coupling of Majorana modes was primarily proposed by the Leiden group.

25 - Page 26, "the cleanness of the wires is paramount to implement robust topological quantum computation". This may not be true depending on the protocol implemented. For instance, use of Coulomb coupling-based protocols may be resilient to spurious Majorana modes, see for instance https://journals.aps.org/prb/abstract/10.1103/PhysRevB.88.155435. Similar arguments may hold for other proposals as well.

  • validity: ok
  • significance: low
  • originality: low
  • clarity: ok
  • formatting: excellent
  • grammar: excellent

Author:  Ville Lahtinen  on 2017-07-26  [id 157]

(in reply to Report 1 on 2017-06-02)

We would like to thank the referee for his/her careful reading of our manuscript and the numerous suggestions that help us improve it. It is highly appreciated to receive such a detailed high quality report.

The main criticism, summarised also by the editor, concerns the lack of clear focus group. After reading the manuscript now with fresh eyes, we agree that the level of presentation is not always consistent with the aimed audience of undergraduates / early post-docs with solid understanding of quantum mechanics and the basics of quantum information. We have addressed this by expanding the introduction where now the intended audience and the level of discussion is explained. We also give references to all the other books / reviews / lecture notes on topological quantum computation that we are aware of in order to place our review in context. In the main text we have dropped all the unnecessary condensed matter topics, while explaining better at conceptual level those that are retained.

The list of major changes can be found in the attached file. The specific points raised by the referee are addressed below.

Yours truly, Ville Lahtinen and Jiannis K. Pachos

——————————————————————————— Requested changes and our replies ———————————————————————————

1 - In the introduction, Sec. 1, the Authors manage to avoid mentioning two fundamental concepts behind the idea of topological quantum computation, adiabaticity and degeneracy. While these are discussed later, I think the two concepts are so important (while also being understandable to anyone with a good basic knowledge of quantum mechanics) that it would be beneficial to mention their crucial role topological quantum computation.

Thank you for this suggestion. It is indeed a good idea to emphasise these two concepts from the very beginning. We have now added a sentence in the first paragraph to state these important concepts explicitly.

2 - Page 2, "(at least what comes to our every day energy scales)". This is a rather obscure remark, not sure what is meant by that.

Unfortunately, our attempt to be a bit more casual made it obscure. We agree that it is not an important remark and that it can be confusing. The remark has now been removed.

3 - The paragraph below Eq. 3 is also quite obscure for the uninitiated reader. First, the concept of fractionalization is introduced here first without any explanation. It is linked immediately to strong interactions, but the connection is not clear. This is an example of a paragraph which would be unintelligible for someone without the proper condensed matter background, for instance a knowledge of fractional quantum Hall effect. Also, note Majoranas do not require strong interactions, yet they are the most treated example in the rest of the work; they are also not so easily linked to fractionalization or the Aharonov-Bohm effect. So it is also not clear that the concept quickly introduced here are beneficial to the reader to understand the material that follows.

Our aim here was to guide the reader's thinking with a simple example towards viewing anyons in general requiring quite special conditions. We agree though that immediately talking about strong interactions can be both confusing and misleading. We have revised the last paragraph of Sec 1.1 to explain better the motivation of this discussion.

4 - In the paragraph below Eq. 3, the sentence "This happens if the charge or the flux is fractionalised, i.e. Q \neq n (in units of the electron charge) or \Phi \neq 2\pi n, respectively" is unclear. First, Eq. 3 is only a joint condition on the product q\Phi. Also, physical units for flux are off, or a definition of the flux quantum is missing.

We have revised the sentences such that it is clear that only either the charge or the flux needs to be fractionalized. We also now explicitly specify that the quantum of flux equals 2\pi.

5 - The first two paragraphs of Sec. 1.3 are too dense and can not be followed without knowing several advanced concepts of condensed matter physics. At this point in the manuscript a definition or description of topological order has not been attempted yet (not even a heuristic one), yet these paragraphs dwell on the distinction between SPTs and intrinsic topological order. Again, it's not clear what the benefit is for the reader. The role of topology in condensed matter is very broad and diverse and it is not always directly relevant for TQC, especially in a supposedly gentle introduction such as this one.

Our motivation here was to convey to the reader that topological order does not in most cases necessarily imply the presence of anyons. We agree though that the level of discussion on the topic was unnecessarily tight for a newcomer to follow. We clarified this discussion.

We have extensively revised the corresponding paragraphs to clearly spell out the difference between SPT and intrinsic topological order. We also further included sentences describing how defects (with examples what kind of defects there can be) can also bind anyonic zero modes in SPT states, even if they do not support anyonic quasiparticles as intrinsic excitations.

6 - Last sentence of page 5, "topological insulator heterostructures". This term seem to exclude semiconductor heterostructures, which are actually an even most prominent candidate to engineer topological superconductivity. I would suggest to reformulate the wording to make the term more encompassing of all the different proposals.

Absolutely true. The sentence has been reformulated to treat both approaches on equal footing.

7 - Page 6, first paragraph: "off-diagonal conductance" is a confusing and unusual term for the familiar Hall conductance.

Indeed. We have now changed it to Hall conductance.

8 - Page 6, "Employing the intricate connection to conformal field theory,...". Which connection? No mention of such connection was made so far in the text. This sentence comes out of the blue and, again, is very obscure for someone not knowledgeable about the topic.

We agree once again. The sentence brings no extra information in this context and we have removed it from the revised manuscript.

9 - Page 7, "... it has been known that if the pairing in a 2D superconductor is so-called p-wave type, i.e. time reversal is broken by the pairing term, then..." Breaking TRS is a necessary but not sufficient condition to obtain p-wave pairing, hence this sentence is slightly misleading. This passage should be expanded not to make things sound more simple than they are.

Indeed. We have corrected the sentence to clearly say that both are independently needed.

10 - Page 7, last paragraph before Sec. II. It is not clear from this paragraph if these quantum simulations are only equivalent to topologically ordered states via a non-local mapping, or if they realize the ordered state directly. Judging from the list of references, it probably conflates the two things, which conceptually is not helpful.

Cold atom systems do realise the actual topological states, while the references on cavity arrays and photonic systems are indeed only equivalent via non-local mapping. In the spirit of not introducing vague concepts, we prefer not to spell this out explicitly, but we have reordered the sentences such that actual topological states are treated separately from the non-locally equivalent ones. We also added a remark the emphasis that the latter are not strictly speaking topological states, but refrain from talking about non-local mappings to avoid having to explain what they are.

11 - In Sec. 2, it would be nice to give some more information about pentagon and hexagon equations. These are mentioned several times but never really discussed, not even at a heuristic level, so a sense of mystery is left in the reader. As a consequence, some passage are obscure. For instance, why do the hexagon equation have more than one solution, and to the pentagon equation also share this property?

It is our opinion that the role played by pentagon and hexagon equations is worth mentioning as means of obtaining consistent anyon models. However, as they do not feature in quantum computation itself, it is not worth digging deeper into their solutions as the data they give can be looked up in the given references. To be more clear about this, we know explicitly mention that their role lies in the classification of anyon models and we have removed sentences relating to details, such as the number of solutions.

12 - Sec. 2.4: "Unlike Fibonacci anyons, the fusion space of Ising anyons has a natural tensor product structure.". This sentence is crucial but also mysterious, because the strucutre of the Hilbert space of Fibonacci anyons was not discussed in the section regarding Fibonacci anyons. It should be explained how the Hilbert space of Fibonacci anyons has no tensor product structure, and why this is special. The same problem features in a similar sentence in the first paragraph of Sec. 3.1. The structure of the fusion space of Fibonacci anyons is invoked twice without prior discussion of the notion.

We have added discussion to the section on Fibonacci anyons to clearly state what the lack of tensor product structure means and why it poses a caveat for performing quantum computation.

13 - Page 13, second paragraph, "means that it is also protected against any uncorrelated local perturbations." What does uncorrelated mean here? Is it a redundant adjective or is it essential? Do "correlated" local operations do something in the fusion space?

Here it is actually redundant and has now been removed. It was meant to capture correlations in local perturbations giving rise to effective non-local perturbations, but since in the next sentence we explicitly talk about unlikely non-local perturbations, that remark was redundant.

14 - After Eq. (21), "While this is likely to be challenging in the laboratory..." I would say it is more likely to be very noisy, than challenging. Such a coupling can be easily achieved at least in the case of Coulomb-coupled Majorana modes.

Agreed. Implementing such coupling is not the challenge, but getting to phase correct is. We added a side sentence to explicitly mention this.

15 - Page 18, "In the presence of thermally excited anyons the statistics of anyons as is expected to become ill-defined and..." First, this sentence is grammatically broken. Second, it is not clear why the statistics of anyons becomes ill-defined.

We have rewritten the whole paragraph and removed this sentence. It is a too vague explanation why temperature poses a problem to encoding quantum information in intrinsic topological states. We now only refer to the references to point out that it is likely to have a problem when encoding quantum information in intrinsic topological states with finite temperature. We also point out that anyons bound to defects in SPT states are likely to be more robust.

16 - Sec. IV, first paragraphs. The use of the adjective "massive" is counterproductive here. First, this word has many heuristic meanings depending on the context. Second, I think there is no easy sense in which Majorana zero modes are "massive". What is missing here is a thorough discussion of the distinction between anyons as true quasiparticles (as in the FQHE) or as quasiaparticles associated with topological defects. So again, without further elaborations (which may go beyond the scope of the paper) the use of the term is confusing.

In the revised manuscript we now already in Sec 2.1 clarify the distinction between anyons as defects in SPT states and as excitations in intrinsically topologically ordered states. We have now revised this paragraph such that we explicitly talk only of excitations in intrinsic states where anyons are massive excitation and mention explicitly that a finite-amount of energy input is needed to create them.

To our understanding Majorana zero modes can also be said to be massive in a microscopic system in the sense that the defect binding them carries mass. For instance, this is the case with vortices in topological superconductors. We agree though that grinding on these subtleties is likely to lead only to confusion and hence we refer now only to intrinsic excitations.

17 - The first pargraph of Sec.4.2 needs serious improvement: first, "the degeneracy of the ground state in the presence of anyonic quasiparticle excitations" is an oxymoron (ground state in presence of excitations); second, anyons may not arise as true excitations of the system (topological superconductors have no anyonic excitations in the strict sense of the term); third, ground state property and properties of the excitations are mixed throughout the paragraph.

Yes, this was indeed very poorly phrased for an uninitiated audience. We have revised the paragraph to be clear that is the lowest energy state in the presence of anyons that exhibits the protected degenerate subspace. We are now also careful to make it clear that the concepts of topological entanglement entropy and degeneracy apply only to ground states of intrinsic topological states.

18 - "The degeneracy equals the number of distinct anyons (including the vacuum)." This statement is given without a heuristic explanation, and as such it is not very helpful and becomes quite forgettable.

In the spirit of cutting unnecessary concepts, in the revised manuscript we no longer talk how either the topological degeneracy or the topological entanglement entropy depend on the details of a particular anyon model.

19 - "The degeneracy is most conveniently obtained from the Verlinde formula [124]." This is another concept/formula which is name-dropped without explanation or clear benefit to the reader.

We have revised to make this section much more colloquial and have removed many such more technical statements in order to keep the discussion simple. All the discussion related to how the degeneracy depends on the anyon model and the topology of the surface has been removed.

20 - Entanglement entropy is also never defined, not even heuristically. It is quite an advanced concept to assume a beginner reader to be familiar with. Same problem with "quantum dimension" and "topological twists" later in this section.

Basic definition of entanglement entropy is now included to explain how the scaling is obtained. At the same time, we removed any direct discussion how the correction relates to particular anyon model and just give references for that.

21 - Page 21, "Since the work of Ivanov, ..." While of course Ivanov's paper is a seminal one, I think the braiding properties of vortices in p+ip superconductor where also discussed in Read & Green.

True. We have added the reference to Read & Green also here.

22 - Page 22, "...implies that Majorana fermions could...". I feel that by now Majorana fermion is a discouraged term in this context. Also, terminology should be consistent and it seems in the rest of the work authors prefer the more precise "Majorana zero modes".

Corrected. Our intention was to always use “Majorana mode”, but this had escaped our proof-reading.

23 - Page 22, "It took 10 years to discover that Kitaev’s simple model could be realized by depositing a spin-orbit coupled semiconducting nanowire on top of a normal s-wave superconductor [56, 57]." I would mention the necessary presence of a magnetic field.

We now explicitly mention this.

24 - Page 25, "These include switching domains of the wires between Coulomb energy and charging energy dominated regimes enabling Majorana modes to jump between different domain walls [137, 138]". This sentence is wrong, since charging and Coulomb energy dominated regimes are the same thing. The Authors probably meant charging vs Josephson regimes. Also, the References are wrongly placed here; the Coulomb coupling of Majorana modes was primarily proposed by the Leiden group.

This is indeed a stupid typo and we have corrected it. We have also revisited the references and cited them accordingly.

25 - Page 26, "the cleanness of the wires is paramount to implement robust topological quantum computation". This may not be true depending on the protocol implemented. For instance, use of Coulomb coupling-based protocols may be resilient to spurious Majorana modes, see for instance https://journals.aps.org/prb/abstract/10.1103/PhysRevB.88.155435. Similar arguments may hold for other proposals as well.

We have added the reference regarding this possibility.

Attachment:

changes_MaQYSe6.pdf

---

## Round 1 · Referee Report · Thomas O'Brien (Referee 2) · 2017-6-8

Strengths

1- Good introductory survey of the field of topological quantum computing for a non-expert

2- Covers appropriate definitions and examples (Fibonacci and Ising anyons) without going into unnecessary detail.

3- Gives complete non-technical description of how to use anyons to perform quantum computation (at least to the point of Ising anyons), up to the level of detailing Majorana wires in the Kitaev model.

Weaknesses

Nil.

Report

This is a well-written review article that was a pleasure to read, and a nice addition to the existing literature for non-experts wishing to learn more about topological quantum computing. Though other similar reviews do exist (Freedman et. al., Nayak et. al., Trebts et. al., as well as lecture notes by Roy and DiVincenzo and by Preskill, and books by Wang and by Pachos), this distinguishes itself by being short, non-technical, and having a specific focus on anyon computation rather than specific physical realizations.

As it stands, none of the requested changes below are critical, but instead comments to further improve the manuscript. As it is aimed at an audience not well-versed in the field, I feel that readibility and accessibility is key, and hope that these suggestions can help with this.

Requested changes

1- In general, as this review is for non-experts in the field, it would benefit from more interconnection between the various sections. For example, Sec.5.1-5.2 gives a physical model which realizes the Ising anyons of Sec.2.4; a forward reference to the latter section in Sec.2.4 would help a more physically-minded reader who is finding it difficult to grasp the notion of Fusion channels.

Similarly, in Sec.2.1 the fusion channels are described as the only options for the order in which anyons are to be fused. A naive reader might ask why in Fig.3 particles a and c cannot first be fused. Tying this section together with Sec.2.2 to answer why this case is not considered (because such a fusion would first require braiding of a and b or c and b) would probably be of assistance to the reader.

Also, the 'sigma anyons' in Sec.4.1 should probably carry a reference to Sec.2.4 for a reader who has skipped this example.

Also, a reference to Sec.5 in the last paragraph of Sec.2 where experimental realizations of Ising anyons exist might be useful.

2- In the definition of fusion channels (Sec.2.1), the authors assert that the Fusion rules give rise to a basis for the Hilbert space in which a topological quantum computation would be performed. Although this is unequivocally true, it is very non-trivial and somewhat counter-intuitive to a new reader (after all, fusion is a process occurring over a span of time, whilst a Hilbert space is usually associated with a system at a fixed point in time). The nature of the basis vectors is described at (in appropriate detail) in later sections (in particular 5.2.1), but it might be preferable to either link to this in the earlier section, or give a longer description of what a fusion space describes physically (this difficult given the fairly detailed maths underpinning this section, but if a few sentences with a reference were added here it would add much insight to a key part of the text).

3- Section 4.2 describes the link between quantum dimension and entanglement entropy, which gives means by which the anyon model for a given physical system can be partially understood from entropy scaling. However, the purpose of this section is not made clear in the first few paragraphs, which instead discuss long-range correlations and the physical system topology. At this point in the review, this discussion is a slight diversion, and the entire section is somewhat difficult to follow. I would suggest dropping the second paragraph (which could potentially be moved to a separate section and expanded on, but this is entirely optional), and combining the first and third paragraphs to immediately draw the reader's attention to equation (26) and the relationship $\gamma=\log \mathcal{D}_M$, as these are the key points of this section.

4- In Sec.1.1, the fact that one particle encircling the other is equivalent to two exchanges is somewhat non-trivial. I do not know a good way of explicitly demonstrating this without using diagrams (i.e. showing a deformation of the two exchanges to a loop). But perhaps the sentence below equation 1 can be rewritten to describe this in more detail. The word 'encircle' here should probably be replaced by 'pass' as well. On a related note, it is somewhat difficult to see that the line in the 3D paths of Fig.1 should go behind the right-hand particle; perhaps this should be noted in the caption?

5- Recent work on Majorana modes in 2D heterostructures (theory: https://arxiv.org/abs/1608.08769 and https://arxiv.org/abs/1609.09482 , expt: https://arxiv.org/abs/1705.05049) seems somewhat promising and should probably be mentioned along with the wire experiments.

6- I would switch Sec.2.4 and Sec.2.3, as Ising anyons are a bit easier to understand as a first example than Fibonacci anyons.

7- In Sec.2.4, below equation 14, the Hilbert space dimension is only given for systems with 2N Ising anyons, whilst the example above uses an odd number thereof. It should probably be explained that Ising anyons only come in pairs, and that the example given in eq (13) considers a subsystem where an additional anyon would also be present.

8- Sec.5.2.2 could include some more details on proposals for universal quantum computation in Majoranas, as well as perhaps some more details of the evolution of the quantum state whilst Fig.10 evolves, so as to make better connection between the braiding and the underlying physical model.

9- In equation 3, it could be noted that if $2q\phi=2\pi n$ the same evolution occurs, but it is bosonic/fermionic.

10- When describing the classification scheme for systems that support anyons in Sec.1.2, a sentence on how general this scheme is would be useful.

11- There are a very large number of grammar errors throughout the paper. Below I have written down a list of those I noticed whilst reading, however it is quite possible that I missed many myself, and so it would be advisable to further run the text through a grammar editor.

Abstract, s.4: 'general steps how to use' -> 'general steps for using', 'as well as discuss' -> 'as well as discussing the', 'various ways' -> 'various ways that'.

Sec.1: par.1, s.3: 'statistical behaviour' -> 'statistical behaviour,'

Sec.1.1: par.2, s.1: 'However, in the real world small definitions matter.' par.2, s.6: 'error in implementing quantum gates' -> 'erroneous manipulation of the physical system'. par.4, s.2: 'it dictates' -> 'this dictates'. par.7, s.1: 'is not equivalent to actually existing' -> 'is not equivalent to it actually existing'. par.8, s.1: 'The study of the strongly correlated system' -> 'The study of strongly correlated systems'. par.8, s.3: 'This enables to' -> 'This enables us to'

Sec.1.2: par.1, s.2: 'It is possible to characterise them' -> 'It is possible to characterise this distinction', 'enable to classify' -> 'enable a classification of'. par.1, s.3: 'topological classification is only defined given that' -> 'topological classification requires only that'. par.1, s.7: 'but due to the particle-hole symmetry' -> 'but due to particle-hole symmetry'. par.1, s.9: 'the most experimentally accessible ones' -> 'the most experimentally accessible anyons'. par.5 s.1: 'FQH states can also emerge' -> 'FQH states can emerge' par.6, s.4: 'study physically most common' -> 'study (the physically most common)' par.6, s.5: 'could exist [34]' -> 'could exist [34],' par.9, s.1: '(iii)' -> '(iv)'. par.9, s.3: 'cold atoms [68]' -> 'cold atoms [68],'

Sec.2: par.1, s.1: 'what are their defining properties' -> 'what their defining properties are'

Sec.2.1: par.2, s.1: 'fusion channel degree of freedom enables to define' -> 'fusion channel degree of freedom enables the definition of' par.3, s.6: 'set of the fusion matrices' -> 'set of fusion matrices', 'the pentagon equations, the $F$-matrices as they are often called,' -> 'the pentagon equations (often called $F$-matrices)'

Sec.2.2: par.1, s.2: 'In particular, anyons are exchanged, or braided as the process is often called due to their worldlines forming braids.' -> 'In particular, anyons are exchanged or braided (as the process is often called due to their worldlines forming braids).' par.1 s.9: 'Thus when there is fusion space degeneracy' -> 'Thus, when there is a fusion space degeneracy', 'of the exchanges but' -> 'of the exchanges, but'.

Sec.2.3: par.2, s.5: 'enable to test and develop' -> 'enable testing and development of'

Sec.2.4: par.1, s.2: 'implies that when brought together two' -> 'implies that, when brought together, two' par.1, s.2: 'the last fusion rule which' -> 'the last fusion rule, which'. last par, last sentence: 'be operated and then Section 5 illustrate' -> 'be operated, and then in Section 5 we illustrate'

Sec.3: par.2, last sentence: 'present in laboratory' -> 'present in a laboratory'.

Sec.3.1: par.2, last sentence: 'choose a computational basis as the pairwise fusion basis' -> 'choose the pairwise fusion basis as a computational basis'.

Sec.4.2: par.2, last sentence: 'can not unambiguously' -> 'can not be unambiguously'.

Sec. 5.2.1: par.1, s.3: 'braiding statistics is only defined projectively, i.e. up to the overall phase' -> 'braiding statistics are only defined projectively, i.e. up to an overall phase'

Sec. 6: par.3, s.1: 'More realistic route' -> 'A more realistic route'.

  • validity: high
  • significance: good
  • originality: ok
  • clarity: high
  • formatting: good
  • grammar: acceptable

Author:  Ville Lahtinen  on 2017-07-26  [id 156]

(in reply to Report 2 by Thomas O'Brien on 2017-06-08)

Dear Mr. O’Brien,

Thank you very much for taking the time to carefully read though our manuscript and produce a high level report. We highly appreciate the kind words as well as the numerous comments that help us improve our manuscript.

In the light of the suggested changes, as well as those in the other reports, we have carried out extensive revisions. These changes aim to improve the readability and interconnectedness of the different sections, explain all new concepts briefly, but concisely, as well as clarify the intended audience of the review. We have also carried out a thorough proof-reading to catch typos and have added all the suggested further references.

The list of changes can be found in the attached file.

Best regards,
Ville Lahtinen and Jiannis Pachos

Attachment:

changes.pdf

---

## Round 2 · Referee Report · Anonymous (Referee 1) · 2017-8-14

Strengths

1 - Readability.
2 - Length.
3 - Good entry point to the field and to the rest of the literature.

Weaknesses

I find that the weaknesses I indicated in my first report have been successfully addressed by the Authors.

Report

I read the new version and, in particular, went through the list of changes submitted by the Authors. I thank the Authors for considering the remarks that popped up in the first round of reviews. I find that the manuscript has considerably improved in the new version. The text has increased by a few pages, but this has allowed more breath and clarity in the exposition. The concepts that were not properly introduced before are now explained or at least briefly defined with pointers to the literature.

Overall, I think this review work is ready for publication and that it will be a valuable addition to the literature in general and to the journal in particular. I have one minor revision to be considered before the article is published, see below.

Requested changes

10 - In the last paragraph before Sec. II, I would emphasize more that the proposals in Refs. 81-83 are qualitatively different than those in the preceding references. While interesting, I don't think these works can lead to a platform to topological quantum computation, since as also mentioned by the Authors they do not really realize topological systems. Hence, they are sort of misplaced here. If the Authors want to refrain from explaining Jordan-Wigner transformations or other non-local mappings - a sentiment which I agree with - they can still state that these proposals are only unitarily equivalent to a system with anyons, in a way which is not protected by local errors and is thus non-topological in nature.

  • validity: good
  • significance: good
  • originality: ok
  • clarity: high
  • formatting: excellent
  • grammar: excellent

Author:  Ville Lahtinen  on 2017-08-15  [id 161]

(in reply to Report 1 on 2017-08-14)

Thank you very much again to carefully go through our manuscript and helping us to improve. We are happy to hear that our revisions were satisfactory and we have now also addressed the final remark according to the suggestion. Details can be found in the list of changes in the resubmission.

---

## Round 2 · Author Response

Thank you very much for the high quality refereeing process.

According to the editorial recommendation, we have carried out extensive revisions to clarify the intended audience and to make our more work accessible for them. We also decided to slightly change the title in order to avoid confusion, but also to make connection with the book by Pachos, where many of the condensed matter topics are discussed in more detail. The list of major changes can be found attached.

Sincerely yours,
Ville Lahtinen and Jiannis K. Pachos

---

## Round 2 · List of Changes

List of changes

Title:
We changed the title to “A Short Introduction to Topological Quantum Computation” in order to have a unique title that is not confused with Pachos's book.

Abstract:
We have revised the abstract to more clearly communicate the contents and the intended audience.

Contents:
After deliberating on the optimal structure of the review, we decided move Section 5 (“Manifestations of anyons...”) to be a subsection of Section 2. In this way it does not interrupt the flow of the discussion that progresses from anyons models (Sec 3) to quantum computating with them (Sec 4) to the example (Sec 5).

We also revised the titles of the Sections 5.1-5.5 to better reflect their content.

Section 1:
As suggested by referee 1, we have added a sentence to the first paragraph to immediately introduce the key concepts of a degenerate protected subspace and adiabatic transport to manipulate the states within it.

We have also added two new paragraphs. The first explains the intended audience of the present review and puts our work in the context of other reviews/books on topological quantum computation. The second new paragraph explains the structure of the review.

Section 2:
We have completely revised the three paragraphs before Sec 2.1. Without going deeper into the mathematics, our intention here is to make clear the difference between SPT order and intrinsic topological order and clarify that the first requires defects of some type to support anyons, while the latter does not. In the third paragraph we also now anticipate the concepts of topological entanglement entropy and topological degeneracy that arise in intrinsically ordered systems and which are explained in Section 2.2.

Section 2.2:
This section is the old Section 5. The first paragraph explains the scope and purpose of the section.

Section 2.2.1:
Throughout this section we make it clear that when talking about excitations we refer to intrinsically ordered systems where anyons are massive excitations, but that similar manifestations of protected degeneracies apply also to SPT states with defects. We also try to anticipate the discussion ahead and connect the protected degenerate subspaces and Berry phases to encoding and processing of quantum information that is described in the following sections.

Section 2.2.2:
This section has now been shortened and we explicitly mention that the concepts of topological entanglement entropy and topological degeneracy only apply to intrinsic topological states. Regarding the nature of topological degeneracy, we no longer discuss how it depends on the anyon model, but only give references. After all, our intention is just to introduce the reader to these two concepts that appear often in the literature, but that do not directly feature in topological quantum computation.

In the same spirit, we now define how entanglement entropy of a system is calculated, but omit the details how it depends on the anyon model via the total quantum dimension. We also no longer talk of “topological twists” when extracting the full data.

We also made the decision to drop the paragraph related to edge states and their relation entanglement spectrum in order to cut unnecessary concepts that are not directly relevant to quantum computation.

Section 3.1:
Regarding pentagon and hexagon equations, we now mention explicitly that their role is to classify possible consistent anyon models, but refrain from talking about them any deeper since their solutions can be looked up from literature for all anyon models of interest. We don’t feel that discussing them in depth provides any significant understanding of anyons from the point of view of quantum computation.

Section 3.3:
We now give as eq. (16) the fusion rules for few different numbers of Fibonacci anyons to show how the dimensionality of the fusion space grows and why it implies a lack of tensor product structure.

To clarify this key difference to Ising anyons that do have a tensor product structure, we also added the fusion rules for many Ising anyons as eq. (18).

Section 4.3:
We have revised the paragraphs discussing the challenge posed by finite temperature to topological quantum computation. We omit any discussion about “ill-defined statistics” and merely state the results from the given references that discuss Abelian anyon based quantum memories. Any discussion about temperature in nanowires is deferred to Section 5.4.

Section 5.2:
We have clarified how the microscopic properties of the wire, such as fermion parity conservation, relate to the Ising anyon model and how they are used in the protected encoding of the Majorana qubit.

Section 5.3:
We thought how to expand and provide more details about the calculation of braid evolutions in the wire network, but in the end decided that providing any of the mathematics behind the braiding calculations does not serve the purpose here. Our aim is merely to justify that topological quantum computation is indeed possible in nanowire arrays along the principles outlined in Section 4. We have revised the section though the make all our statements clearer and we now also explicitly point to references where details how braiding and measurement is carried in a particular realization of nanowire arrays could be carried out.

Furthermore, we added a paragraph to describe how the general measurement protocol of Section 4.3 is to be carried out in nanowire arrays.

Further revisions:
Proof-reading to catch typos

We have added all the suggested references as well as few other that we came across.

Improved inter-connectedness by referring back and forth to different sections.

---

## Round 3 · Author Response

Thank you very much again for the high quality reviewing process with our first submission to SciPost. We have addressed the final remark by the referee (details in the list of changes) and hope that the manuscript is now ready for publication. We also added few more references to recent works that were brought to our attention.

Best regards,
Ville Lahtinen and Jiannis K. Pachos.

---

## Round 3 · List of Changes

Last paragraph of Sec 2.1:
We have split the paragraph into two paragraphs, where the first deals with simulations of actual topologically ordered systems, while the second deals with those that are only unitarily equivalent. As suggested by the referee, we now explicitly mention that unitary equivalence does not imply topological protection and mention that the aim of this approach is to simulate and test control operations in an encoding that parallels that of a genuine topological encoding.

We also added references [91,92,93] on simplified ways to construct braids for topological quantum computation, as well as reference on contemporary review on Majorana modes in solid-state systems [75] and reference [165] on a recent blueprint for a measurement-only topological quantum computer.

---

## Editorial Decision

published